# Intelligent wearable olfactory interface for latency-free mixed reality and fast olfactory enhancement

Yiming Liu [1,2,13], Shengxin Jia [1,3,13], Chun Ki Yiu[1,3,13], Wooyoung Park [1,13], Zhenlin Chen[1,3], Jin Nan[4], Xingcan Huang [1], Hongting Chen[2], Wenyang Li[1], Yuyu Gao[1], Weike Song[5], Tomoyuki Yokota [2,6], Takao Someya [2,7,8] ✉, Zhao Zhao [5] ✉, Yuhang Li [4,9,10,11] ✉ & Xinge Yu [1,3,12] ✉

Olfaction feedback systems could be utilized to stimulate human emotion, increase alertness, provide clinical therapy, and establish immersive virtual environments. Currently, the reported olfaction feedback technologies still face a host of formidable challenges, including human perceivable delay in odor manipulation, unwieldy dimensions, and limited number of odor supplies. Herein, we report a general strategy to solve these problems, which associates with a wearable, high-performance olfactory interface based on miniaturized odor generators (OGs) with advanced artificial intelligence (AI) algorithms. The OGs serve as the core technology of the intelligent olfactory interface, which exhibit milestone advances in millisecond-level response time, milliwatt-scale power consumption, and the miniaturized size. Empowered by robust AI algorithms, the olfactory interface shows its great potentials in latency-free mixed reality (MR) and fast olfaction enhancement, thereby establishing a bridge between electronics and users for broad applications ranging from entertainment, to education, to medical treatment, and to human machine interfaces.

Olfaction, as an ancient evolutionarily critical physiologic system in humans, plays a significant role in our interaction with the surroundings, including but not limited to detecting potential hazards in the environment, shaping social behavior, smoothing negative emotions, and recalling buried memories[1]. Despite the enormous influence of olfaction on human life, a handful of olfaction-generating technologies (also called olfactory interfaces) have been developed for olfactory display in some potential applications, ranging from human–machine interactions, to education, to entertainment, and to clinical treatment[2–4]. Compared to the extensive research works on the development of haptic feedback technologies[5–13], olfactory-generating technologies have obtained little attention in the recent years, resulting in long-term technical stagnation. Considering the natural defects of massive olfaction generators in overall size, response time and

[1]Department of Biomedical Engineering, City University of Hong Kong, Kowloong Tong, Hong Kong, China. [2]Department of Electrical Engineering and Information systems, The University of Tokyo, Tokyo 113-8656, Japan. [3]Hong Kong Center for Cerebra-Cardiovascular Health Engineering, Hong Kong Science Park, New Territories, 999077 Hong Kong, China. [4]Institute of Solid Mechanics, Beihang University, 100191 Beijing, China. [5]China Special Equipment Inspection and Research Institute, 100029 Beijing, China. [6]Institution of Engineering Innovation, The University of Tokyo, Tokyo 113-8656, Japan. [7]RIKEN Center for Emergent Matter Science (CEMS), Saitama 351-0198, Japan. [8]Thin-film Device Laboratory, RIKEN, Saitama 351-0198, Japan. [9]Tianmushan Laboratory, NA, 311115 Hangzhou, China. [10]Aircraft and Propulsion Laboratory, Ningbo Institute of Technology Beihang University (BUAA), 315100 Ningbo, China. [11]Liaoning Academy of Materials, NA, 110167 Shenyang, China. [12]Hong Kong Institute for Clean Energy (HKICE), City University of Hong Kong, Hong Kong, China. [13]These authors contributed equally: Yiming Liu, Shengxin Jia, Chun Ki Yiu, Wooyoung Park. ✉e-mail: someya@ee.t.u-tokyo.ac.jp; zhaozhao@csei.org.cn; liyuhang@buaa.edu.cn; xingeyu@cityu.edu.hk

power consumption, the development trend of next-generation olfactory interfaces is approaching wearable formats with miniaturized design to achieve rapid olfaction feedback and generate localized odorous environment[2,14,15]. Up to now, the developed wearable olfactory interface systems are mainly built on arrays of odor generators (OGs) for supplying multiple odors, where the working principles of the OGs variate from the piezoelectric-based atomizer[16] to the controllable phase change of the odorous wax[17]. However, both two types of OGs require second-level response time, resulting in an obvious latency for users during practical applications, where the response time of the OGs is defined as the delay time in generating or terminating odors. For example, Metaverse, as one of the most important olfactory interface application fields, is demanded to provide a smooth, immediate, and immersive experience to users, requiring an low delay time from both human–machine interface system and network.

Apart from the challenges in the latency issue of OGs, a number of technical difficulties still exist, including miniaturizing the size, optimizing the power consumption, and building up high channel counts/density[18]. For instance, the state of art odors generating methods generally require a whole set of bulky equipment with limited odors supplied and professional operation, forcing users to visit a specific place to experience olfactory feedback every time, which is unfriendly to users[19]. So, olfactory interfaces simultaneously exhibiting sub-second response time, centimeter-scale size, tens of OGs array, and micro-watt low power consumption is challenging (Supplementary Table 1)[2,3,16–18,20–22].

Here, we report a series of materials, algorithms, devices, mechanics, electronics, and integration strategies for AI-driven, wireless olfactory interfaces, which exhibit world-record high response time (0.07 s), OG array density (0.75 unit/cm$^2$) and latency-free feature. The miniaturized OGs used in the olfactory array adopt air flow and heat as odor-generating factors as a working principle, where the release of odors is controlled by a mechanical actuator to open/close the breathing holes in the OG, while the odor concentration is controlled by the tuning of heating temperature (Fig. 1a and Supplementary Figs. 1, 2). The combination of advanced AI algorithms with the olfactory interface allows zero latency for the olfactory interface in mixed reality (MR) applications, which enables efficient olfaction training for patients who are experiencing olfactory degeneration. The results show the great potential of the technology in wide applications, ranging from human–machine interactions, to medical treatment, remote education, and to entertainment.

## Results

### Concepts of the olfactory interface

Figure 1a shows the schematic diagram of the odor generator, which is the core component of the miniaturized olfactory interface system. The OG adopts a layout of multilayer stacking (Fig. 1a and Supplementary Fig. 1): a top polyethylene terephthalate (PET) layer (0.1 mm thick) with a 4 × 3 breathing hole array serves as the odor release outlet, and odorous additive could be injected inside OGs for filling the exhausted ones through the breathing holes (Supplementary Fig. 3); a mechanical actuation structure, consisting of the magnet coin (0.5 mm in thickness) and an underlying copper (Cu) coil (300 turns, 50 μm in diameter of the Cu wire), is used to control the PET cantilever up and down inside the OG by variating the current directions into Cu coil. This design could significantly improve the mechanical stability of the OG, especially when users do intense activities; a layer of cotton/adhesive (0.1 mm in thickness) works as the odorous chemicals container, where the chemicals could be in either liquid or powder formats; a layer of the polyimide (PI, 2 μm thick), a metallic electrode composed of gold (Au, 200 nm), chromium (Cr, 40 nm), and a tiny thermistor (1 mm × 0.5 mm × 0.5 mm) act as a heating platform with controllable operation temperature (from 35 to 55 °C, shown in

Supplementary Fig. 4), therefore, controlling the odor generation rate; a 3-dimension (3D) printed square ring (epoxy) supports the whole the structure against the unpredictable external impact. The OG adopts a solenoid valve structure to majorly control the odor release rate, where the breathing holes at the top PET layer could be opened and closed by manipulating the electromagnetic field (Fig. 1a–c). For operating the OGs, an external powering management system provides two channels of voltage inputs (1.7 V and 9.0 V) to respectively supply the electromagnetic coil and heating electrode for controlling the magnet motion and heating temperature insides. To monitor the heating temperature, the miniaturized thermistor is connected to an Analog-to-Digital Converter and General-purpose input/outputs of the microcontroller in a self-developed control panel for continuously measuring the temperature (sensing resistance) variation during the operation. According to the real-time heating temperature, the control panel could rapidly turn on/off the power supply into the heating electrode for stabilizing the temperature at the target one. To change the duty cycles, amplitude and frequency of the alternating current (AC) power into the electromagnetic coil, the control panel adopts the decoder controlled by two digital General-purpose input/outputs to convert external direct current (DC) power source to AC power for realizing a programmable magnetic up-and-down displacement (Supplementary Notes 1, 2).

Figure 1d, and Movies S1 show that the simulated equivalent strains in the electrode remain lower than the elastic limit (0.3%) in the external distributed pressure of 8.8 kPa. When the magnet is driven by the uniformly distributed load to bend the cantilever structure downwards by 29°, the PET is the first part of the whole structure to reach its elastic limit (0.4%), as shown in Fig. 1e and Movies S2. The core technical parameters of the OG reported in this work show significant improvement, compared to the state-of-the-art ones, including the response time in controlling odor release, the lowest operating power consumption, overall size, stability of repeatedly switching the heating temperatures, minimum operation temperature, and maximum number of available odor types in one olfaction interface (Fig. 1f)[2,4]. Due to the excellent performances of the OGs, the olfaction interfaces based on it can be utilized in various scenarios ranging from MR-based entertainment, education, olfaction recovery, and enhancement (Fig. 1g).

### Electrical properties of the odor generator

Figure 2 shows the result of a series of experimental optimizations performed on the breathing hole patterns, the internal heating temperature, the frequency, amplitude and duty cycle of the power input into the electromagnetic coil, from which we can find that the optimized parameters enable the OG to exhibit ultra-fast response time, precise odor generation control, and low-power consumption. Supplementary Fig. 5 presents the effect of breathing hole patterns in the top PET layer on the performance of the OGs, where five OGs adopt five different breathing hole patterns with different solid area ratios (0.914–0.996) of the PET layer (Supplementary Figs. 5a–c). Supplementary Figure 5d, e shows the ethanol concentration generated by the five OGs as a function of breathing hole patterns, and it is obvious that the OG#1 with the lowest solid area ratio (0.914) could contribute to the highest stabilized ethanol concentration (~2649 ppm). To further investigate the effect of the breathing hole patterns on the electrical property of the OGs, we recorded the volunteers' sensed odor intensities to 32 different odor types (see details in Characteristics) generated by the five OGs (Supplementary Figs. 5f, g), where there are 6 males and 4 females in the volunteer test. It is found that the average odor intensity scores of OG#1 is the highest among the five OGs, consistent with the results shown in Supplementary Figs. 5d, e. Therefore, the first breathing hole pattern (Supplementary Fig. 5b) is adopted. Figure 2a, b presents the heating temperature of OGs as a function of the running time at a fixed heating power, duty cycle and

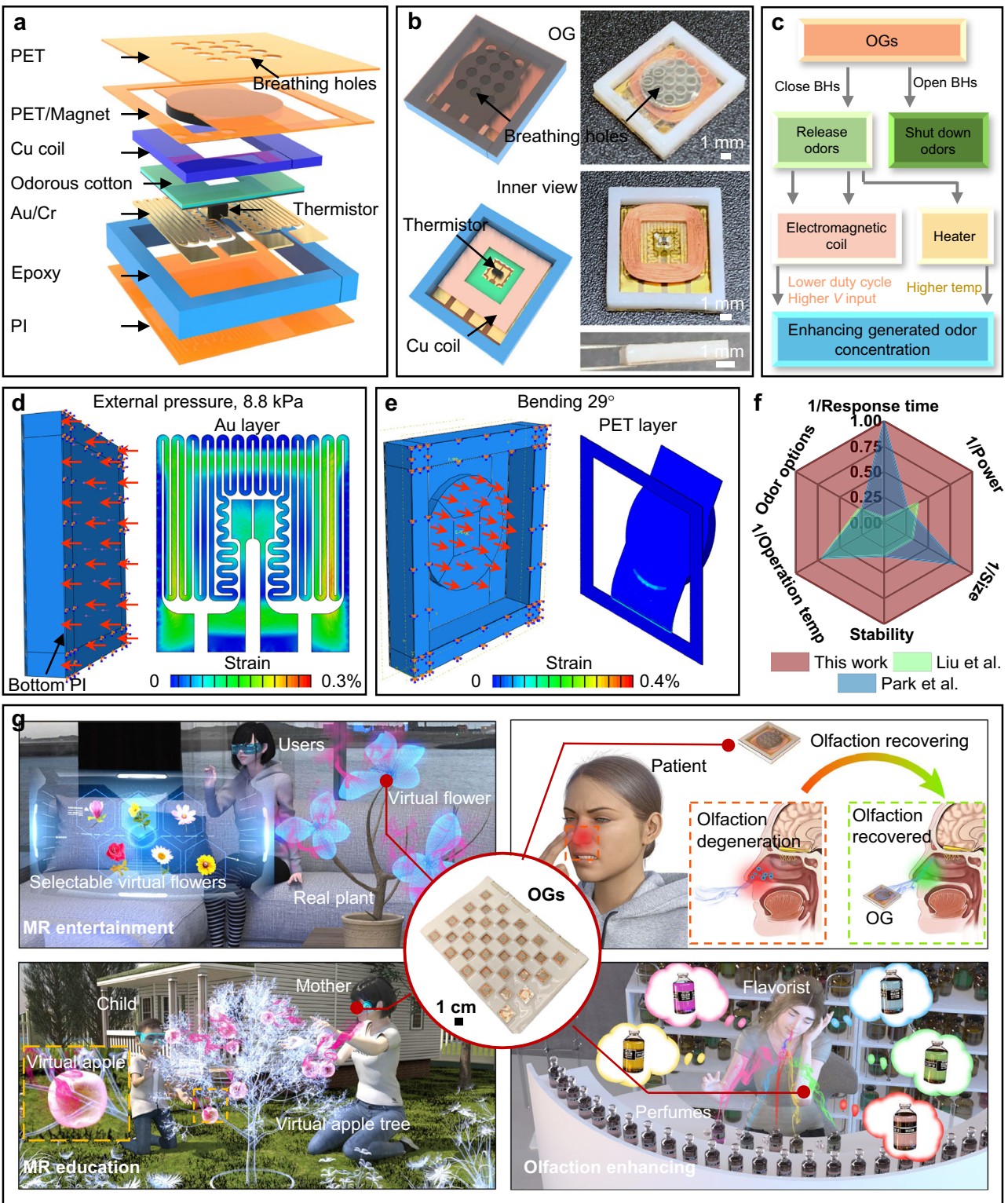

**Fig. 1 | Concept of the olfaction interfaces. a** Exploded view of the odor generator with detailed descriptions of each component. **b** Optical images and schematic diagram of the OG with the additional inner views. **c** Manipulation methods of the OGs in controlling odor generation rate. **d**, **e** Mechanical stimulations of the Au-based heating electrode (**d**) and cantilever-like PET film (**e**) under the distributed external pressure and bending deformation, respectively. **f** Comparison of the OGs and the recently reported ones in terms of response time, lowest power consumption in operating OGs, overall size, stability, operation temperature, and the maximum number of supply odors. **g** Schematic diagram of the advanced OG-based olfaction interfaces for providing olfaction feedback to users at four typical applications, ranging from MR-based entertainment and education to olfaction training for olfactory recovery and enhancement.

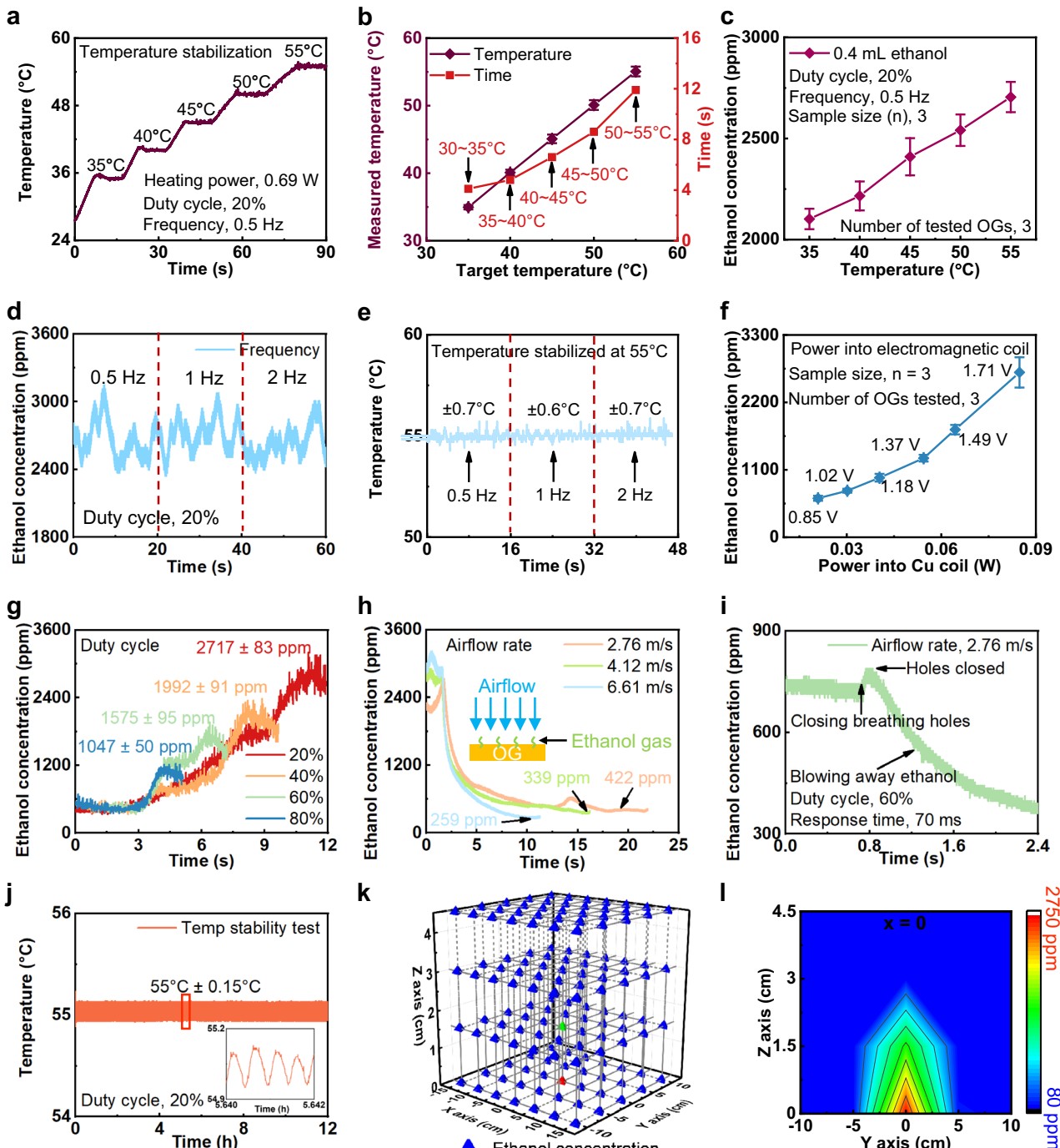

**Fig. 2 | Optimal electrical performance of the OGs. a** Heating temperature of OGs as a function of operation time with the target temperature variating from 35 to 55 °C. **b** Response time required to reach the target temperature with the measured stabilized temperature shown in (a). **c** Stabilized ethanol concentration generated by OGs as a function of heating temperature ranging from 35 to 55 °C. Here, the error bars are presented as the standard deviation, and the test was repeated 3 times by testing 3 different OGs. **d, e)** Stabilized ethanol concentration (d) generated by OGs as a function of frequency of AC power into the electromagnetic coil with the corresponding heating temperature variation (**e**). **f** Stabilized ethanol concentration as a function of amplitude of AC power into the electromagnetic coils. Here, the error bars are presented as the standard deviation, and the test was repeated 3 times by testing 3 different OGs. **g** Stabilized ethanol concentration as a function of AC power duty cycle, including 20%, 40%, 60%, and 80%. **h)** Ethanol concentration variations along with the operation time at the three different surrounding airflow rates, including 2.76 m/s, 4.12 m/s, and 6.61 m/s. **i)** Changes in ethanol concentration at the surrounding airflow of 2.76 m/s when the breathing holes are closed by manipulating the movement of magnet inside OGs. **j** Heating temperature variations along with the 12-h operation time with the target temperature of 55 °C. **k, l** 3D distributions of the stabilized ethanol concentration above OGs with the detailed 2D concentration at $x = 0$ mapped in l.

frequency of 0.69 W, 20%, and 0.5 Hz, respectively. The results show that it takes 4.1 s, 4.8 s, 6.6 s, 8.6 s, and 11.9 s to gradually increase the heating temperature from 30 to 55 °C at an interval of 5 °C. The minor temperature fluctuations ranging from 0.33 °C @ 35 °C to 0.72 °C @ 55 °C have proven the excellent electrical performance of the OGs in

accurately controlling heating temperature. Figure 2c shows the positive linear relationship between the peak ethanol concentration (odor release concentration) and the OG heating temperature, increasing from 35 °C @ 2102 ppm to 55 °C @ 2705 ppm. Supplementary Fig. 6 presents the operation time of the three OGs with the

constant 0.04 mL ethanol embedded as a function of the heating temperature ranging from 55 °C to 35 °C, where 80 ppm is the human olfactory threshold value to ethanol[23]. Here, the operation time of the OG is defined as the duration when the OG starts to continuously release odor until that the odor intensity above the breathing holes of the OG is lower than the corresponding human olfactory perception threshold value. It is obvious that the higher heating temperature could result in a shorter operation time (Supplementary Fig. 6c), which may result from the more intense Brownian motion and air convection induced by larger temperature differences between the heating electrode and ambient air temperature. Considering the available commercial ethanol sensor (TGS823, Figaro Engineering Inc.) to detect the odor concertation, we chose the ethanol solution (vol%, 95%) as the odorous chemical stored inside OGs instead of other biosafe odorous chemicals. Although it is accessible to control odor generation intensity by variating the heating temperature, it also results in a second-scale response time (>4 s) caused by increase the heating temperature. Therefore, during practical applications, it's necessary to explore and combine other techniques to control the generated odor concentration for the purpose of shorter response time.

Instead of the heating temperature controlling method, we also investigated the effect of frequency, amplitude, and duty cycle of AC power into the electromagnetic coil on the generated odor concentration in Fig. 2d–i. Figure 2d and Supplementary Fig. 7 show the stabilized ethanol concentration as a function of AC power frequencies (0.5 Hz, 1 Hz, and 2 Hz) with a constant heating temperature, power amplitude and duty cycle of 55 °C, 84.8 mW and 20%. It is obvious that the peak ethanol concentrations at the three frequencies are stabilized around 2717 ppm with minor differences. Furthermore, we recorded the heating temperature along with the operation time at the three AC power frequencies, as shown in Fig. 2e. The three uniform heating temperature fluctuations ($\pm$ 0.7 °C @ 0.5 Hz, $\pm$ 0.6 °C @ 1 Hz, and $\pm$ 0.7 °C @ 2 Hz) and same peak ethanol concentrations at the three frequencies demonstrate that the frequency of the AC power has no significant influence on the peak odor concentration. Figure 2f shows that the generated ethanol concentration increases with the amplitude of AC power into the electromagnetic coil from 20.9 mW @ 635 ppm to 84.8 mW @ 2689 ppm. The result indicates that the amplitude of AC power into the coil could have a positive effect on the odor generation rate, which results from the fact that a higher amplitude of AC power could induce a more intense actuation, contributing to a higher odor diffusion rate. In addition to the AC power amplitude, we also measured the released ethanol concentration by the OGs along with the operation time at the four different duty cycles of the AC power into the coil (20%, 40%, 60%, and 80%), where the 20% duty cycle of the AC power demonstrates that the open time of the breathing holes is four times longer than the closed time (Fig. 2g). As observed, it takes 0.99 s, 3.26 s, 5.09 s, and 7.58 s to reach up to the corresponding stabilized values, 1047 ppm, 1575 ppm, 1992 ppm, and 2717 ppm at the four duty cycles (20%, 40%, 60%, and 80%), proving that higher duty cycles could lead to a higher odor concentration (Fig. 2g).

Considering the fact that the airflow rate induced by users' motion may affect the electrical performance of the OGs, we recorded the released ethanol concentration along with the operation time at the three high airflow rates (2.76 m/s, 4.12 m/s, and 6.61 m/s), where these rates offer sufficient range to cover the most practical application requirement (Fig. 2h). It is clear that a higher airflow rate surrounding OGs could suppress the generated ethanol concentration to a lower stabilized value, such as 422 ppm @ 2.76 m/s, 339 ppm @ 4.12 m/s, and 259 ppm @ 6.61 m/s, where the three ethanol concentration values at the beginning of the test with the airflow rate of ~0 m/s are around 2717 ppm. To assess the response time of OGs in controlling odors, we adopted a commercial ethanol sensor (TGS823, Figaro Engineering Inc.) to respectively monitor the real-time variations of ethanol concentration with the operation time at the surrounding airflow rates of 0 m/s and 2.76 m/s, during which we closed the breathing hole at 0.33 s and 0.72 s (Supplementary Fig. 8 and Fig. 2i). It is clear that the both of two ethanol concentrations are increased slightly in a short time (70 ms), then starts dropping down. The concentration enhancements at the time point when the breathing holes are closed are caused by the internal ethanol gas ejected from the OGs, induced by the upward motion of the magnet. As a result, the response time of the OGs could be drastically reduced to 70 ms. Here, it is worth mentioning that closing the breathing holes of the OGs will block the ethanol release from the OGs, further resulting in a significant decrease in ethanol concentration, and higher ambient air flow rates could induce a faster odor diffusion rate, accelerating the decreasing rate of the ethanol concentration (Supplementary Fig. 8 and Fig. 2h, i).

Supplementary Fig. 9 shows the generated ethanol gas concentration by the OGs as a function of the ethanol concentration in the solution inside the OGs at the constant duty cycle of AC power into the electromagnetic coil and heating temperature of 20%, and 55 °C, and the result indicates that higher ethanol concentration in the solution could contribute to a higher generated ethanol gas concentration. Therefore, it is preferred to select strong perfumes as the odorous additives within OGs. Figure 2j shows the heating temperature response of the OG at the target temperature 55 °C. The minor temperature fluctuation ($\pm 0.15$ °C) and 21600 working cycles in opening and closing the breathing holes by the electromagnetic effect demonstrate the high stability of OGs with the enlarged details in Fig. 2j. During the experiments, the ethanol commercial sensor is uniformly located on the breathing holes of OGs for monitoring the odor concentration variations in Fig. 2c, d, f–i as the larger distance between the sensor and breathing holes will result in a lower odor concentration in air (Fig. 2k, l), which is induced by the fast diffusion rate of ethanol gas in open air. To investigate the gas tightness of the OG, we monitored the ethanol concentration around the OG with the breathing holes closed, during which the heating temperature of the OG is programmed to increase from 35 °C to 55 °C to assess the heating temperature effect on the gas leakage from the OG (Supplementary Fig. 10). It is found that the gas leakage is observed with a maximum ethanol concentration reaching up to 255 ppm at the heating temperature and distance between the OG and ethanol sensor of 55 °C and 0 cm, respectively. The leaked ethanol concentration (255 ppm) is slightly lower than the human minimum identifiable odor level (350 ppm)[24], indicating that although the leaked alcohol concentration can be perceived by humans, it cannot be identified by humans due to the low concentration. When the distance between the OG and sensor is set at 1.5 cm, the leaked ethanol concentration ranges from 49 ppm to 62 ppm, which is out of human perception range, where the ethanol concentration at the distance of 1.5 cm during the odor-releasing status could reach up to 1491 ppm (Fig. 2l). As a result, users are recommended to experience the odors generated by the olfactory interface at a distance between OGs and users' nose over 1.5 cm to avoid smelling a small amount of leaked odors. Supplementary Figure 11a–j presents a volunteer test with 10 volunteers involved to investigate if human olfactory thresholds are distinguished for 32 different odors. During the test, we recorded the volunteers' reaction time when the volunteers were required to continuously smell the odors at a constant distance between the OGs and the volunteers' noses until they could obviously sense the generated odors. It is interesting to find that the same volunteer may have different olfactory thresholds for different odors, and different volunteers also have distinguished thresholds for the same odor (Supplementary Fig. 11k).

## Latency-free MR demonstration of the olfactory system

Under the umbrella of a self-developed AI algorithm, the olfactory interface based on the OGs could be utilized in MR for realizing latency-free applications. The latency-free olfactory system is not limited to only providing olfactory feedback with zero delay, but can

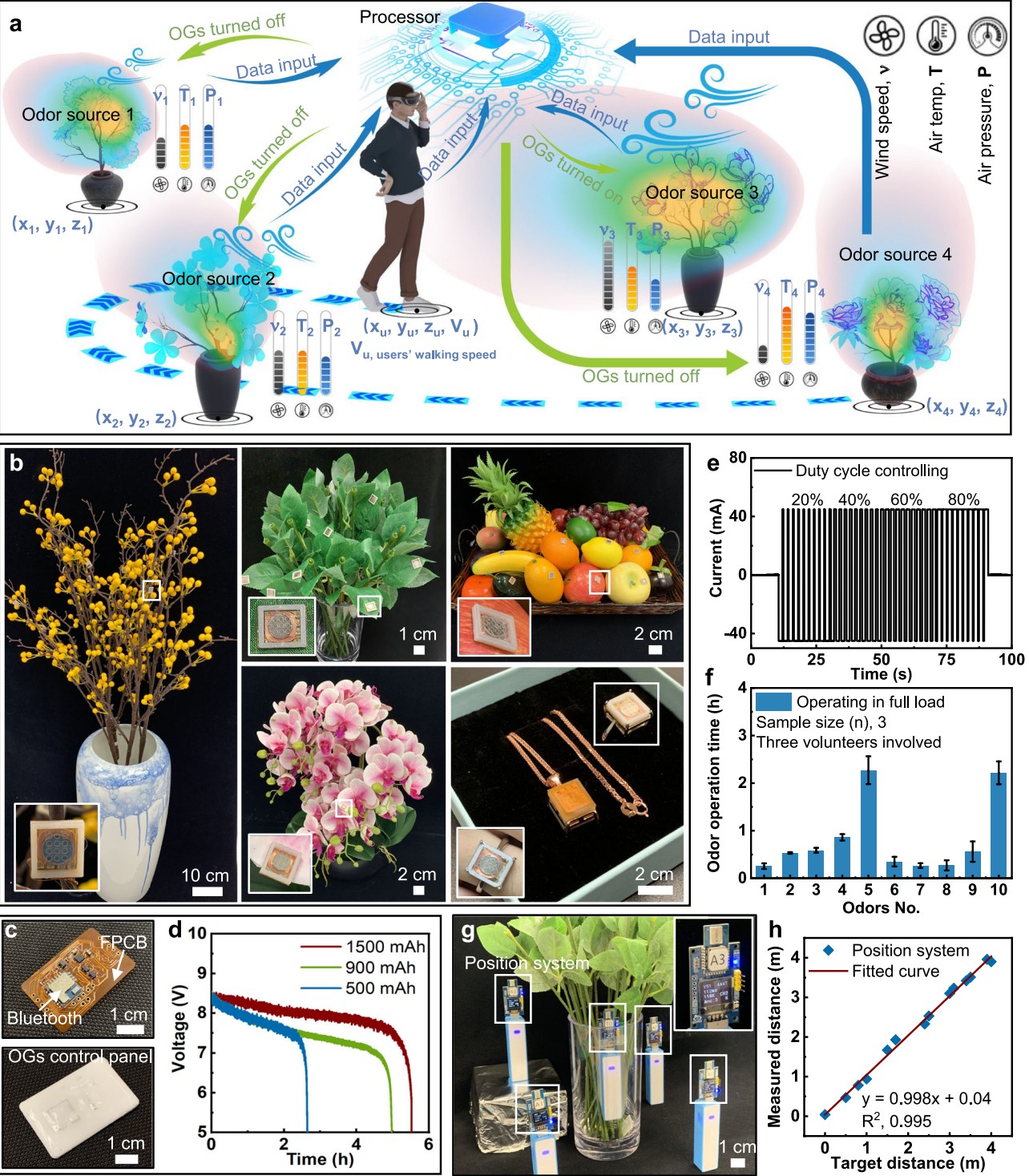

**Fig. 3 | A representative demonstration of intelligent olfaction system for achieving a latency-free MR application. a** Schematic diagram of the olfaction interface integrated MR system for providing a latency-free olfaction feedback to a user by predicting the user's motion for an immersive MR experience, where the analyzed data includes ambient wind speed, air temperature and pressure, user's movement velocity and 3D location, and the 3D locations of odor sources. **b** Optical images of five typical realistic models with olfaction interfaces integrated. **c, d** Optical images (**c**) of the flexible, two-channel control panel with the full-load operation times (**d**) at the three batteries (500 mAh, 900 mAh, and 1500 mAh).

**e** Duty cycle variations into each OG controlled by the 3-channel control panel. **f** Odor generating time of the OGs with 10 different odorants at the full-load operation. Here, the error bars are presented as the standard deviation, and the test was repeated 3 times by testing 3 volunteers. Additionally, the operation time of the odors (No. 11–32) could be found in Supplementary Fig. 20. **g** Optical image of the commercial positioning system with one integrated inside the odor source for real-time measuring its 3D location. **h** Measurement accuracy of the positioning system verified by a standard ruler.

achieve a controllable olfactory feedback delay by setting a delay time in the AI algorithm, thereby simulating the real human olfactory perception in a virtual environment. Figure 3a demonstrates a latency-free olfactory MR concept that a male wearing VR glasses walking around four different odor sources with OGs integrated for generating distinguished odor types, where there is a positioning system, wind speed sensor, air pressure sensor and temperature sensor matched with each odor source for real-time monitoring the 3D location $(x, y, z)$, airflow rate $(v)$, air pressure $(p)$, and ambient temperature $(T)$ around odor sources. Here, the VR glasses are programmed to continuously measure the users' walking velocity $(V_u)$, facing direction, and 3D location $(x_u, y_u, z_u)$. By analyzing all the collected data from the user and environmental conditions, a self-developed algorithm demands the corresponding olfactory interface integrated into the target odor source for operation in advance to compensate for the discrepancy between the user's exploration in the virtual world and its respective odorous environment. In general, odor sources emit odorants in the air where they travel, driven by advection and molecular diffusion on the background of the airflow, which can be turbulent or laminar in nature. The interaction of these transport processes generates the odor landscape, the time-varying spatial-temporal distribution of the odorant concentration in the air[25]. To save energy and odor source, OGs would stay silent until users' possible exploration intent intersects with estimated odor plumes. Based on the user's facing direction and real-time 3D location, the human motion prediction algorithm could calculate the possible users' exploration location, that is long enough for OGs to respond and for odorants to travel from odor sources and achieve an odor-threshold concentration rate at that potential location. Specific to our application, we only consider the average odor concentration when diffusion dominates the flow dynamics in crosswind direction with slow laminar flows. To further simplify the model, we neglected the turbulent airflow and evaluated the time-averaged plume envelope in the odor dispersion process. The odor concentration gradients are calculated using the Gaussian plume dispersion model[26], considering a working OG as a constant release source. In contrast, diffusion coefficients are calculated according to Briggs expression table[27] and atmospheric stability Pasquil classes[28]. (See more details in Materials and Methods).

Figure 3b and Supplementary Figs. 12, 13 present various models with the olfactory interface integrated, where the number of the OGs could be personalized from 1 to 15 for each model according to needs. Here, the olfactory interface supports 32 different common odor types, including fruits, spices, drinks, bakeries, herbs, and others. To minimize the unpredictable artifacts on the olfaction interface in these realistic models, all wire connections are hidden behind the OGs mounting surfaces (Supplementary Fig. 14). In addition to the realistic models with OGs mounted, self-developed flexible control panels based on flexible printed circuit board (FPCB) are developed for wirelessly manipulating the OGs, capable of realizing two-channel OGs independent operation (Fig. 3c and Supplementary Figs. 15–17). The two-channel control panel is powered by a 10 V Lithium-ion battery and supports wireless communication among the olfactory interface, VR equipment (i.e. glasses), and computer (Supplementary Fig. 18). By inputting high and low digital signals alternately to the decoders based on the settled duty cycle, the directions of the current flows into each Cu coil of two controlled OGs are shifted, inducing the directions variations of the electromagnetic forces for manipulating the release or termination of odors (Supplementary Note 1). The flexible format of the control panel allows it to mount on human skin for long-term wearability, where users can also wear it in the necklace and ring models during MR applications.

As mentioned above, the longer response time of OGs could result in a larger diameter of working boundaries for each odor source, inducing the untargeted odor types released when the users walk into the greater overlap region between the target and nearby odor sources. Therefore, duty cycle of AC power into the electromagnetic coil is adopted as the major manipulation method in controlling odor release (Fig. 3e and Supplementary Fig. 19), realized by the control panels shown in Fig. 3c. Figure 3d shows the electrical response of the commercial Lithium-ion batteries as a function of capacities, ranging from 500 mAh to 1500 mAh with the corresponding full-load operation time of 2.7 h, 5 h, and 5.5 h, where the full load operation demonstrates that the heating temperature and AC power amplitude of the two-channel OGs are set at 55 °C and 84.8 mW. As the full-load operation time of the 32 adopted odors ranges from 0.26 to 2.7 hrs on average at a constant volume of 0.04 mL (Fig. 3f and Supplementary Fig. 20), the 500 mAh battery is sufficient for the operation of the olfaction systems in various MR tasks. To enhance the application scenarios, a commercial positioning system (DWM1000, Qorvo, Inc.) is also adopted for continuously locating the odor sources (model/olfaction system), as shown in Fig. 3b, c. The positioning system could be mounted onto different objects in reality to track their 3-dimensional coordinates in real time (Fig. 3g). Here, it is worth mentioning that due to the bulky size of the positioning system, it is difficult to integrate the system with skin-mounted olfactory interfaces, which means that the skin-mounted olfactory interfaces cannot achieve a latency-free MR experience. Following this, the 3D location information is wirelessly transmitted to the computer via Bluetooth mesh. By constructing virtual objects with the same 3D coordinates as those in reality, we could link virtuality and reality to realize the MR experience for users. To verify the accuracy of the positioning system, the measured values of the system are compared with the preset values by ruler, and it is shown that the slope of the fitted curve is 0.998, approaching 1, which proves the high precision of the system (Fig. 3h).

Figure 4 and Movie S3 demonstrate a typical application of the above-mentioned MR system with no-delay olfaction feedback, where the four odor sources (OSs) in reality share the same 3D locations as those in virtuality by utilizing the positioning system (Fig. 4a). Each OS integrated an independent olfaction interface with different numbers of OGs, (OS 1 @ 10 OGs, OS 2 @) 10 OGs, OS 3 @ 15 OGs, and OS 4 @ 10 OGs), capable of providing tens of odor types in one time (Fig. 4a). As a result, users could sense the physical characteristics of the OSs in virtual reality of programmable olfaction feedback. In the demonstration, a male volunteer walked randomly around the four OSs to enjoy the released odors by target OSs with the routine and facing direction (red arrows) shown in Fig. 4b. Figure 4c and Supplementary Fig. 21 sketches a diffusive/laminar regime odor plume structure with smooth odorant concentration. To imitate natural scenarios, we select tea (OS 1), gardenia (OS 2), osmanthus (OS 3), and pineapple (OS 4) to represent evergreen shrubs, woody and herbal flowers, and fruit trays, respectively. For simplicity, we applied 5 ppm as an average odor threshold for all four odor sources and simulated its temporal 3D plume structures with an odorant concentration above 5 ppm. The dispersion of woody flowers (OS 2) is dominated by the environmental wind (avg. 0.92 m/s). Its odor plume can expand to 1.1 m long in about 1.2 s. The other three odor sources' wind speeds always disperse through convective vertical airflow, as OGs can be treated as heat sources while releasing odors. The vertical convective air flows are much slower and take 3.7 s, 3.2 s, and 4.8 s for OS 1, 3, and 4. Their plume structures achieve the height of the volunteer's nose. The heat power of each OG is about 0.77 W at full load, and the temporal distribution is calculated using a numerical integral method with slicing of 2 cm (See more details in Materials and Methods). As shown in the Movie S3 and Fig. 4b, d–g, as a volunteer exploring the MR system, the algorithm evaluates the differences between the potential exploration intents and odor landscapes and ensures when he arrives at the edge of the odor plumes, the odor concentration rate has reached at 5 ppm just in time. By demanding OGs to work at the temporal-spatial crossing point of these two potentials,

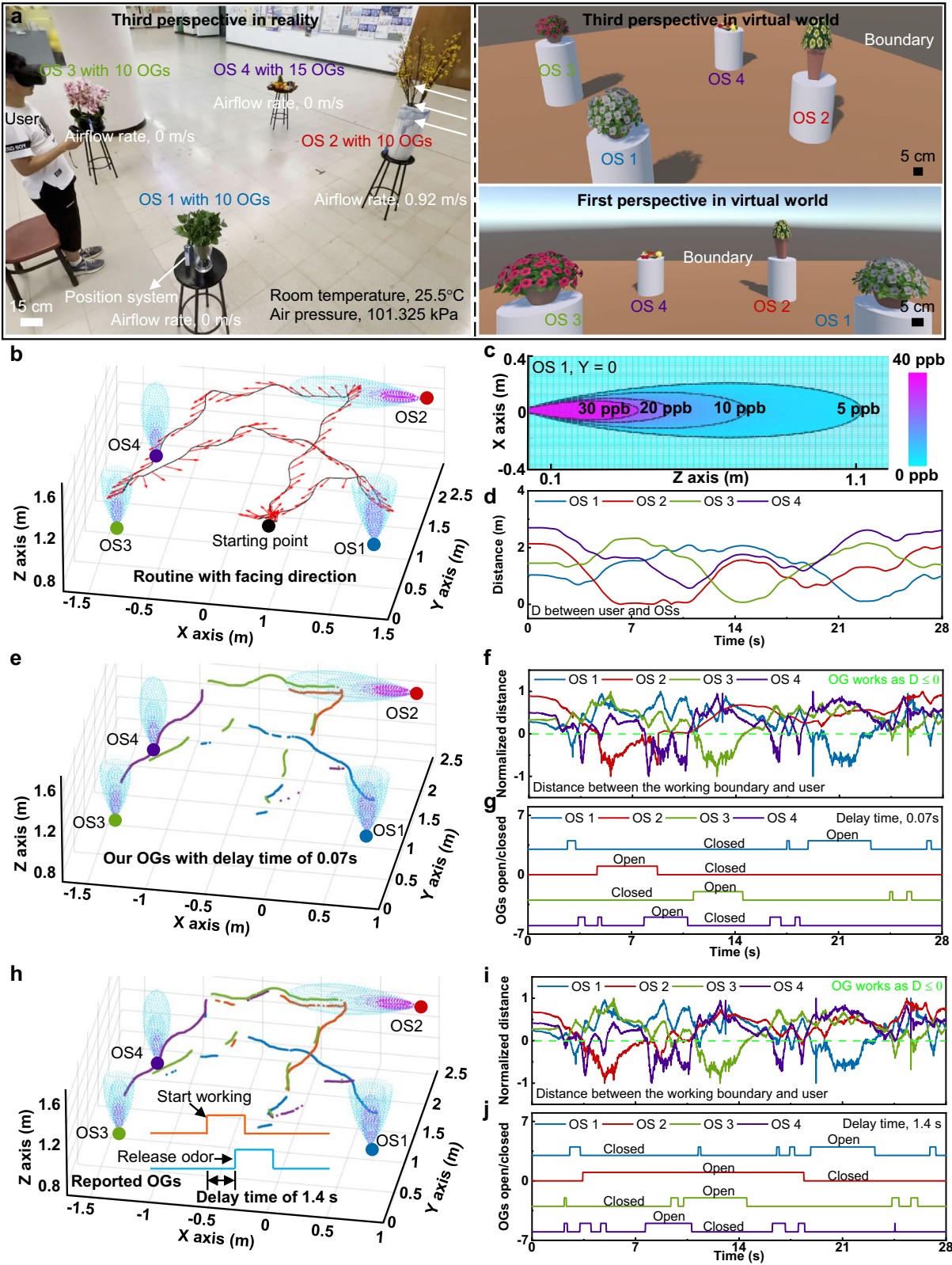

the algorithm achieves latency-free experiences with minimal electrical energy waste. To illustrate the effect of response time, we also simulated the exploration and control process with a 1.4-s response time from the previously reported OG[2]. In contrast to Fig. 4h–j, the operation time of the OG integrated into OS 2 is 4.1 s, approximately one-quarter of that of the previous OG (15.1 s). Therefore, considering the second-level response, bulky size, and high-power consumption,

the conventional odor-generation equipment is not suitable for the latency-free MR applications.

## Olfactory training demonstration of the olfaction system

It is reported that post-upper respiratory tract infection, traumatic, idiopathic and neurological disease-related olfactory disorders can be improved by olfactory training[29]. It is believed that olfactory training

**Fig. 4 | One application of the MR system with the olfaction interfaces integrated. a** Optical images of the whole MR system in reality and virtuality, where a volunteer is walking around the four odor sources for randomly enjoying the target-released odors. **b** The volunteer's exploration trajectory and facing directions within the odor landscape. **c** Plotting of the odor plumes of OS 1 shown in (**b**). **d** Distance from the user's nose towards the downwind axis of each odor sources, $\left|\overrightarrow{UO_p}\right|$. **e** Visualization of command signals sent to the corresponding olfaction interfaces along the volunteer's trajectory. **f** Comparison between the potential distance from odor plumes to the volunteer and the exploration potential, $\widehat{D_{O_pU}} - \widehat{D_{UO_p}}$. **g** Command signals sent to the corresponding olfaction interfaces during volunteer's exploration for achieving a latency-free experience with minimal energy and odorous sources consumption. **h** Stimulation results showing the command signals sent to the corresponding previously reported olfaction interfaces with the response time of 1.4 s along the same volunteer's trajectory. **i** The stimulated comparison between the potential distance from odor plumes to the volunteer and the exploration potential, $\widehat{D_{O_pU}} - \widehat{D_{UO_p}}$, for the reported olfaction interfaces. **j** The stimulated command signals sent to the reported olfaction interfaces during volunteer's exploration for achieving a latency-free experience with minimal energy and odorous sources consumption.

can induce brain neural remodeling and increase the volume of the olfactory bulb, therefore significantly improving the ability of olfactory recognition, discrimination, and the olfactory perception valve[30]. In addition, some studies have found that increasing the types of odors can increase the improvement rate of olfactory function[31]. Therefore, considering the number of integrated OGs, paired power sources, user friendliness, and convenient odorous chemicals refilling, the forearm is selected as the mounting position due to the large skin surface area and high accessibility to human nose. Based on these foundations and remaining challenges, in this demonstration, a flexible, forearm-mounted olfaction interface with 32 OGs embedded is introduced, paired with a self-developed, flexible control panel connected by four bendable flat cables (Fig. 5a, b). Compared to the face mask based olfactory interface with 12 OGs integrated, the forearm-mounted olfactory interface supplies more odor options in a more user-friendly method for a higher efficiency in conducting the olfactory training (Fig. 5b and Supplementary Fig. 22). Here, the gap between OGs is optimized by investigating the permanent magnetic force effect on the motion of the magnets inside the OGs (Supplementary Fig. 23), and it is found that the magnets of OGs could operate as normal once the distance between the center of the magnets is over 1.5 cm. In addition, the corner-to-corner OG layout (Supplementary Fig. 23c) could contribute to a higher OGs array density (0.75 units/cm²) than that (0.55 units/cm²) of the side-by-side layout (Supplementary Figs. 23a, b), as observed in Fig. 5a, b and Supplementary Fig. 22. Therefore, the 32-channel olfactory interface adopts the corner-to-corner OGs layout with a height gap of 2 mm (Supplementary Fig. 23c–e). Both of the two circuit parts adopt a multilayer stacking structure: (1) the top and bottom layers associate with a flexible Polydimethylsiloxane (PDMS, 2 mm, 145 kPa) film as the encapsulation and adhesive layers for providing high resistance against external stimulus and strong adhesion between electronics and mounted skin surface, respectively; and (2) a self-designed thin layer FPCB (Supplementary Fig. 24) interconnects a series of electronic components (a microcontroller unit, a Bluetooth module, resistors, capacitors, a battery, etc.) and 32 OGs for the control panel and olfaction actuation part, respectively (Supplementary Figs. 25–27). To control the heating temperatures and AC power direction into the electromagnetic coils of 32 OGs, a microcontroller (ATMEGA328P-MU, Microchip Technology Inc.) is adopted to control two sets of an array of four shift registers (SN74HC595BRWNR, Texas Instruments Inc.). The 32-channel heating temperatures are measured through four multiplexers (MAX4691-EGE + T, Analog Devices Inc.) and a 24-bit Analog-to-Digital Converter module (ADS1220IRVAR, Texas Instruments Inc.) connected to the microcontroller via Serial Peripheral Interface for higher analog reading resolution (Supplementary Fig. 28 and Note 2). Benefitting from the advanced circuit design, we could accurately control the duty cycle of AC power into each OG and heating temperature wirelessly through the self-developed Graphical User Interface (GUI) installed in a personal computer, shown in Supplementary Figs. 29, 30, and Movies S4, S5. Supplementary Fig. 31 demonstrates a volunteer test (6 males and 4 females) to investigate if any two different odors generated by the 32-channel olfactory interface could be blended into a new odor for users, where 9 different odors (see details in Supplementary Fig. 31) are

randomly selected to create 36 different odor combinations for evaluating the possibility in sensing a new odor. It is obvious that the possibility ranges from 0.19 to 0.64, illustrating a fact that the newly developed high-channel olfactory interface could not stably provide new odors to users by blending the odors embedded as there are many uncertain parameters affecting the mixture of generated odors, for example, the inconsistent distances between 32 OGs and human nose, distinguished odor diffusion rates, and unpredictable ambient air flow rate. Therefore, the 32-channel olfactory interface is programmed to generate one single odor at one time for efficient olfactory training.

To demonstrate the high-channel olfaction interfaces, we called an experimental group with 11 volunteers (5 males and 6 females) and a control group with 14 volunteers (8 males and 6 females) to test human olfactory capability variations with and without 1-hr pretraining time, respectively, in terms of recognition rate and reaction time in distinguishing 32 different odors generated randomly by the olfaction interface (Fig. 5c–g). Here, various olfactory training tests have been designed, encompassing but not confined to: the University of Pennsylvania Smell Identification Test (UPSIT)[32,33], Connecticut Chemosensory Clinical Research Center Test (CCCRC)[34,35], Barcelona Smell Test (BAST-24)[36,37], Snap & Sniff Olfactory Test System[38,39], and "Sniffin' Sticks" Test[40,41]. Although these tests employ distinct protocols and evaluation criteria without a universal gold standard, they are rooted in evaluating users' odor identification or threshold. Therefore, we collected volunteers' recognition rates and reaction time to the 32 different odors for assessing their odor identification and threshold abilities. During the volunteer test, all volunteers were required to wear the 32-channel OGs based olfaction interface system on their forearms, then the experimenter manipulated the self-developed GUI installed in a personal computer for generating a random odor, where the tested volunteer only had one chance to give their answer without time limit for each odor. We refer to the testing method as Random Odor Generation (ROG). It is worth mentioning that all the 11 volunteers in the experimental group had a 1-h pretraining duration before testing their olfactory capability, where each volunteer needed to continuously smell each odor for 2 min in a random sequence. Benefitted from the miniaturized size and ultra-low response time (70 ms) of the OGs, the flexible, high-channel olfactory interface could be directly mounted onto volunteers' forearms (Fig. 5b) and rapidly switch 32 different odor types with imperceptible delay. As a result, the longer pre-training duration could both lead to a higher recognition rate and lower reaction time. For example, the experimental group's average recognition rate is 0.70 after 1-h pre-training, while the control group's recognition rate is only 0.41. In addition to the pre-training duration, we also investigated the training times effect on volunteers' olfaction enhancement (Fig. 5c–g), and it is found that more olfaction training times could contribute to a higher recognition rate and a lower reaction time. For the experimental group, the recognition rate (0.70) @ reaction time (6.2 s) at the third training time, 0.65 @ 8.1 s at the second training time, and 0.59 @ 9.0 s at the first training time. Thereby, both of training times and pre-training duration have a positive effect on the human olfactory enhancement, and the conclusion is consistent with the previous reports[2], further illustrating the excellent practicability of our high-channel olfaction interface. As

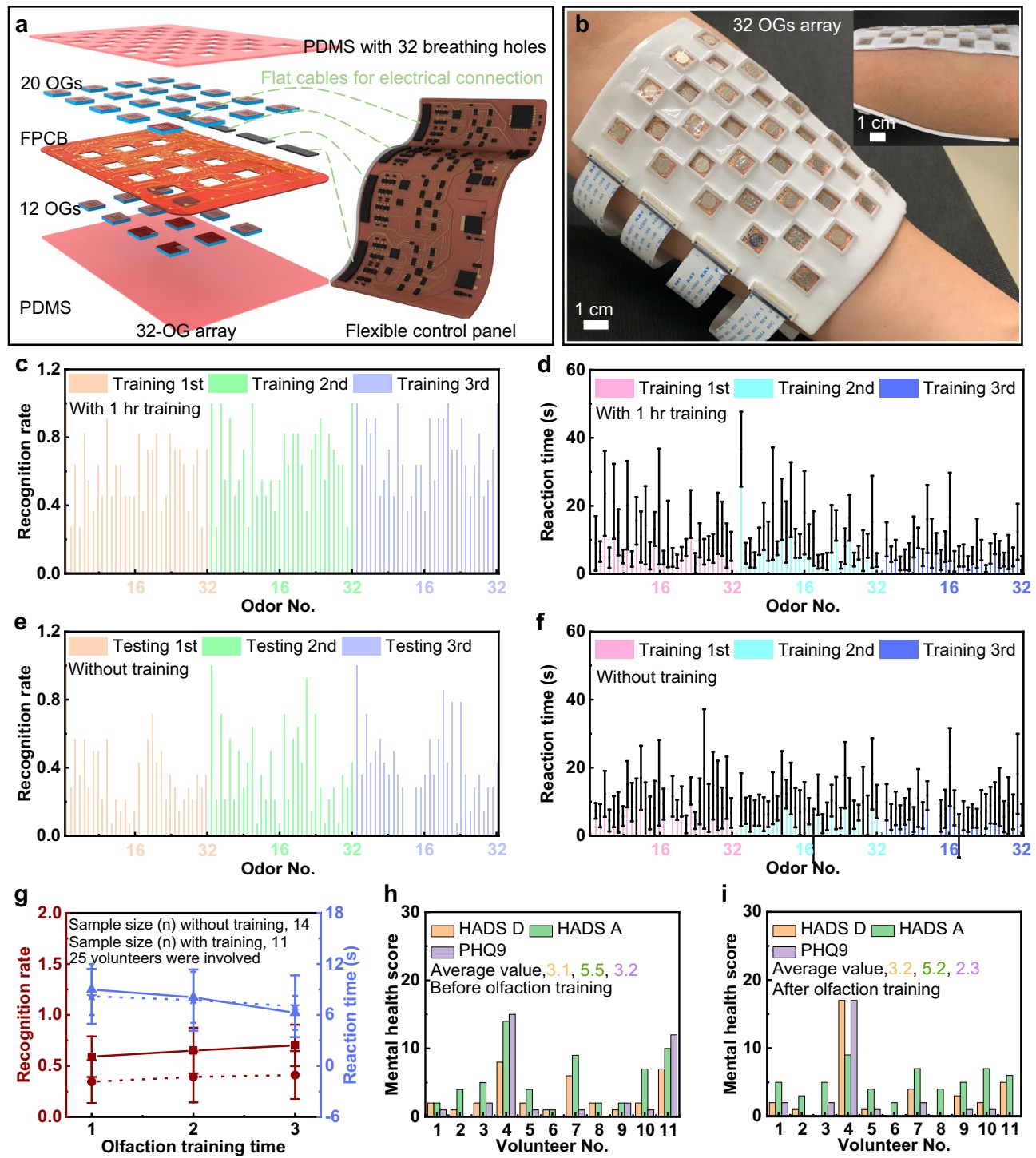

**Fig. 5 | One typical demonstration of the wearable, 32-channel OGs array-based olfaction interface for olfaction training. a** Exploded view of the ultrahigh channel olfaction interface system, consist of two parts, 32-OG array and the corresponding flexible control panel. **b** Optical image of the olfaction interface mounted onto human forearm. **c, d** 32-odor recognition rate (**c**) and corresponding reaction time (**d**) of the volunteers tested three times by the wearable olfaction interface without 1-hr pre-training. Here, the error bars are presented as the first quartile. **e, f** 32-odor recognition rate (**e**) and corresponding reaction time (**f**) of the volunteers tested three times by the wearable olfaction interface with 1-h pre-training. Here, the error bars are presented as the first quartile. **g** Overall results of volunteers' olfactory capability as a function of pre-training time. Here, solid and dotted lines, respectively, represent the data of experimental and control groups shown in (**c–f**). Here, the error bars are presented as the standard deviation, and the volunteer test was repeated 75 times by testing 25 different volunteers (3 times for each volunteer). **h, i** Mental health scores of the volunteers in the experimental group before (**h**) and after (**i**) the three olfaction training times.

olfaction variations can also influence human emotions, therefore, we collected the mental health scores from the 11 volunteers of the experimental group before and after the olfaction enhancement training by adopting two standard emotion assessment forms of Patient Health Questionnaire (PHQ-9) and Hospital Anxiety and Depress Scale (HADS), as shown in Fig. 5h, i. Lower scores of both forms correspond to more positive emotions. The results show that the value of HADS depression (HADS D) variated a little from 3.1 to 3.2

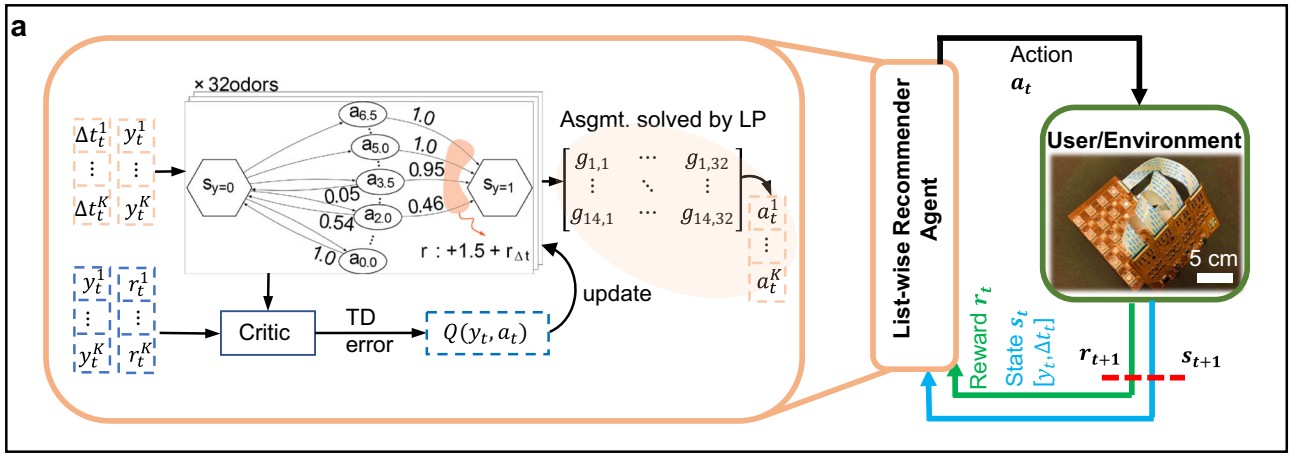

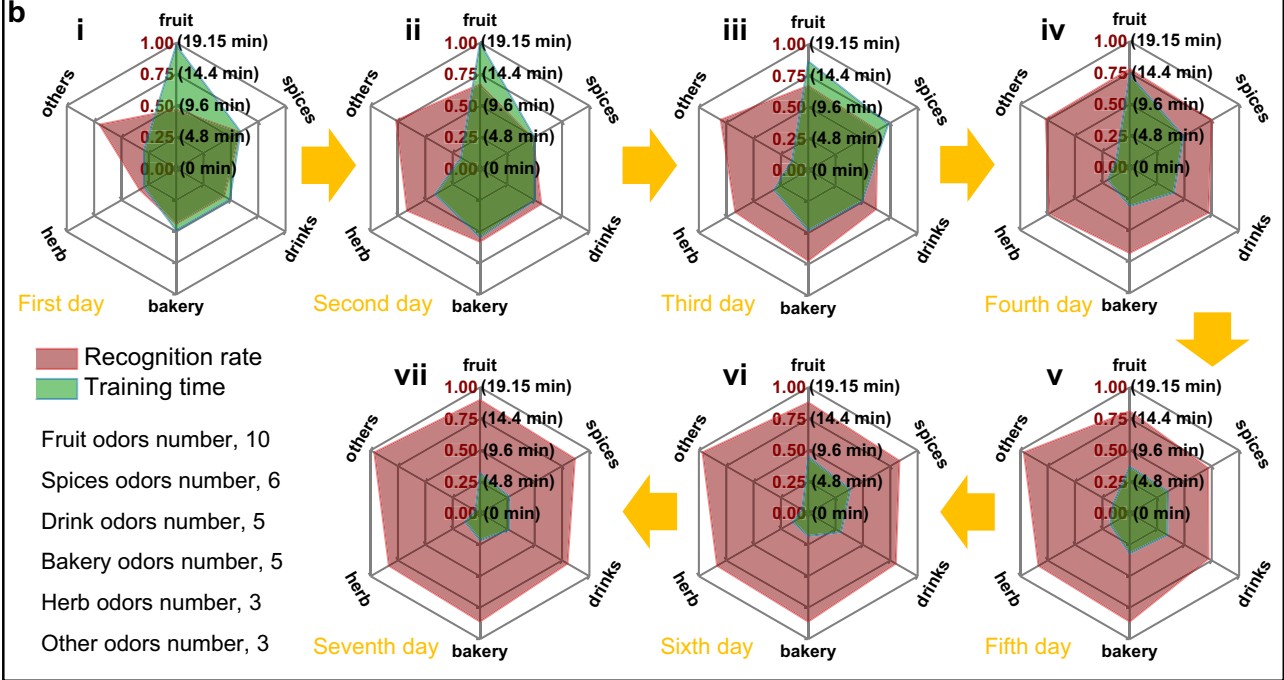

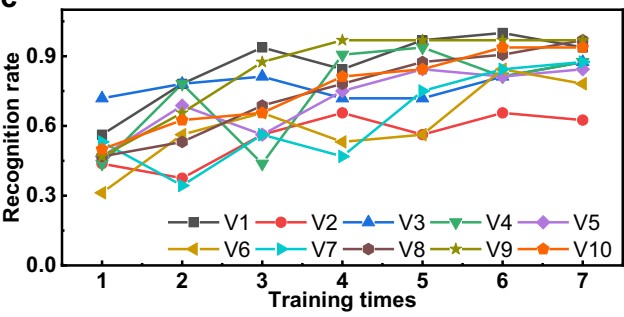

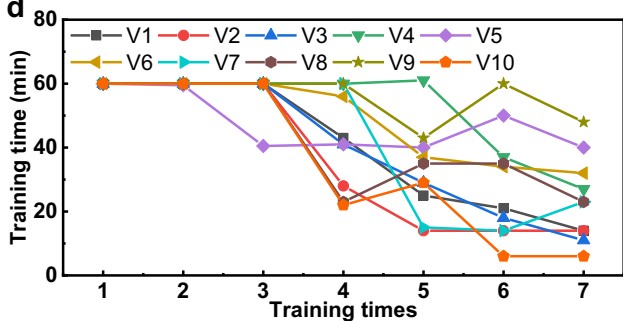

**Fig. 6 | One typical application of the AI-driven olfaction interface in olfaction enhancement. a** The adopted list-wise recommendation frame for providing a personalized olfaction training plan to users. **b** Olfactory capability variations of 10 volunteers without any olfactory impairment after receiving 7-day personalized

olfaction training achieved by the intelligent, high-channel olfaction interface system. **c, d** Overall recognition rate (**c**) and training time (**d**) variations of the 10 volunteers as a function of olfaction training times.

after olfactory training, but values of HADS A and PHQ-9 changed significantly from 5.5 and 3.2 to 5.2 and 2.3, indicating that the volunteers became more positive after 3 olfaction training sessions.

Considering that the ROG method may not be efficient for each user, we developed an intelligent list-wise recommendation framework (LRF) based on reinforcement learning for the training, capable

of providing a personalized optimal training plan, as shown in Fig. 6a and Supplementary Note 3. The recommender agent interacts with the environment (users) by providing a list of recommended training times for all training tasks over a sequence of time steps while the total time for each training session is limited. Like other sequential decision problems addressed in reinforcement learning, this problem is

modeled as a Markov Decision Process, which can be described as 5 tuples (S, A, R, P, γ, Supplementary Note 4). By interaction with users, the recommender agent recommends training time for each odor in such a way that maximizes the expected cumulative reward (total recognition rate), which includes the delayed rewards (Fig. 6a). We follow the standard assumption that delayed rewards are discounted by a factor of γ per time-step. In addition, our agent values the final recognition rate of all odors more than the cumulative rewards during the training procedure. Thus, we define the reward after the action is taken and the expected discounted future reward as:

$$r_{t+1} = \sum_k r_{t+1}^k \tag{1}$$

$$r_t^k = w_1 y_t^k - w_1 y_{t-1}^k \tag{2}$$

$$
\begin{aligned}
R_t &= r_{t+1} + \gamma r_{t+2} + \gamma^2 r_{t+3} + \ldots \\
&= w_1 \sum_k \sum_{\tau=t}^{T} \gamma^{\tau-t} \left( y_\tau^k - y_{\tau-1}^k \right) \\
&= w_1 \sum_k \gamma^{T-t} y_\tau^k - \sum_{\tau=t-1}^{T-1} (1-\gamma) \gamma^{\tau-t} y_\tau
\end{aligned}
\tag{3}
$$

The optimal recommendation strategy maximizes $R_t$, which underscores the final recognition rate at a large discount factor γ. As an extreme case of γ = 1, the discounted return only considers the final recognition test and neglects the return during the training procedure. To further verify the efficiency of the intelligent recommendation system, we called another 10 volunteers (4 males and 6 females) to experience the AI-driven olfactory training based on the high-channel olfaction interface with the test results shown in Fig. 6b–d. Here, to clearly present the volunteers' olfactory capability, 32 different odors are classified into six categories according to type of odorous sources, including 10 fruits, 6 spices, 5 drinks, 5 bakeries, 3 herbs, and 3 others (see details in Characteristics). Each volunteer conducted 7 olfactory training and testing at a frequency of 1 time per day, where the initial training plan is based on the ROG method. It is obvious that these volunteers' olfactory capabilities are significantly enhanced after training with the recognition rates and training time changing from 0.47 @ 18.75 min (fruits), 0.58 @ 11.25 min (spices), 0.44 @ 9.38 min (drinks), 0.44 @ 9.38 min (bakeries), 0.33 @ 5.62 min (herbs), and 0.7 @ 5.62 min (others) to 0.90 @ 6.10 min, 0.87 @ 5.10 min, 0.80 @ 5.10 min, 0.86 @ 4.10 min, 0.83 @ 2.60 min, and 0.97 @ 0.80 min (Fig. 6b–d). It is worth mentioning that the 10 volunteers' olfactory capability approaches the "saturation" condition on the fifth day with the recognition rate and corresponding training time achieving up to 0.81 @ 7.10 min (fruits), 0.72 @ 6.70 min (spices), 0.72 @ 6.60 min (drinks), 0.86 @ 6.10 min (bakeries), 0.83 @ 3.40 min (herbs), and 0.97 @ 2.90 min (others), demonstrating that the LRF aided olfaction interface is capable of enhancing users' olfactory capability in a short time (Supplementary Fig. 32). Supplementary Figs. 33–35 show the average values variation of the 10 volunteers' recognition rate, training time, and reaction time for all tested odors, obviously, the continuous improvement of recognition rate and the constant decay of training time and reaction time further prove the significant enhancement of their olfactory ability. To present the advance of the LRF over the ROG method, we compared the two groups of volunteers' overall recognition rate with the result shown in Supplementary Fig. 36, where the two groups both adopted 1-h pretraining plan with the only difference in training methods. To draw convincing conclusions, the two groups do not share the same volunteers throughout training. It is clear that the slope value (0.54) of the LRF fitted linear curve is much higher than

that (0.40) of the ROG fitted one, illustrating that the olfactory ability improvement induced by LRF is faster than that by ROG.

To demonstrate the role of the LRF aided high-channel olfaction interface system in olfactory recovery, we recruited a patient infected with the COVID-19 virus for personalized olfaction training, who was suffering from severe anosmia after physical recovery (Supplementary Fig. 37a). The patient experienced the training plan at a frequency of 1 time per day at the first two weeks for fast enhancing his olfactory ability (Supplementary Fig. 37b–d). At the first 2 days (olfactory degeneration stage), the patient's olfaction ability shows little change with the recognition rate fluctuating between 0.25 and 0.19 (Supplementary Fig. 37b), which is further proved by the obvious confusion result shown in Supplementary Fig. 37d. From the 3rd to 6th day, his recognition rate and corresponding reaction time is significantly enhanced from 0.38 @ 19.61 s to 0.72 @ 11.83 s, during which we called this stage as olfaction recovering. From the 7th to 14th day, both his recognition rate and reaction time approach the ultimate values variating from 0.78 @ 9.42 s to 0.97 @ 3.8 s with this stage named as olfaction recovery and enhancement. To obtain the normal level of his olfactory ability, we collected his olfaction testing result without training him at a frequency of 1 time per three weeks, and the results show that his recognition rate and reaction time stabilize between 0.72 @ 14.49 s and 0.63 @ 8.78 s. As a result, the AI-driven high-channel olfaction interface system shows great potentials in aiding patients with anosmia improve or restore their olfaction.

In summary, we developed a series of AI-driven, wearable olfactory interfaces based on high-performance OGs for realizing latency-free MR and fast olfaction recovery applications. As the core technology of the olfaction interfaces, the OGs adopt a solenoid valve structure, supporting three methods in controlling odor generation rate, including the majorly operated duty cycle and amplitude of AC power into electromagnetic coil for rapidly closing or opening the breathing holes, and the inner heating temperature controlled by the heating platform. As a result, the miniaturized OGs exhibit technical advances far beyond the reported ones in terms of millisecond-level response time, milliwatt-scale power consumption, overall size, stability, and ultrahigh number of odor supplies. As a result, the olfactory interfaces could realize both of low system delay time and ultrahigh density and throughput OGs array. Therefore, when combining the olfactory interfaces with our self-developed AI algorithms, the systems are capable of providing latency-free olfactory feedback and personalized olfactory enhancement plan to users in MR and olfactory training applications, respectively. The approach could form the starting point of an information channel based on olfaction in many far-reaching applications including VR/AR/MR-based metaverse, online education, and clinical treatment. Efforts to link the olfaction system to physiology, neuroscience, psychology, and materials science would be promising future research directions.

## Methods
### Fabrication of OGs
The fabrication firstly starts with sequential cleaning of a quartz glass with acetone, alcohol, and deionized water (DI water). By adopting double-sided tape, a thin layer of polyimide (PI) with a thickness of 25 μm was attached to the glass sheet. Next, Au/Cr (200/40 nm) were deposited onto the PI film by magnetron sputtering using a machine (Q150TS, QUORUM) and followed by photolithography and etching to obtain metal patterns in the desired geometries. For the photolithography, a positive photoresist (PR, AZ 5214, AZ Electronic Materials) was spin-coated at 3000 rpm for 30 s, soft-baked on a hot plate at 110 °C for 4 min, and then exposed to ultraviolet light for 5 s. After that, the photoresist coated was developed by a developer (AZ 300 MIF) for 15 s and then, by using acetone and DI water, the PR was rinsed away. After that, spin casting a layer of PI (2 μm, 3000 rpm for 30 s, annealed at 250 °C for 30 min) and then selectively etched by Oxford Plasma-

Therm 790 RIE system (patterns defined by photolithography similar as previous step) at the power of 200 W for 10 min, formed encapsulation layers for Au/Cr traces while exposing the connecting patches for later wired connections. After rinsing the PR away, we could obtain the heating electrode (resistance, 118Ω). Fix a thermistor (QN0402X104F4250FB, 1 mm × 0.5 mm × 0.5 mm, Advanced Materials Electronics Co., Ltd) onto the reserved patches in the middle of the heating electrode by silver paste and glue. A cotton layer is attached onto the heating electrode with the middle square area reserved for the thermistor, then a copper coil is directly attached onto the cotton layer with the interface sealed by instant glue. After the glue cured, fix the sample into a chamber with the back cover and surrounding ring wall of a flat PET film and 3D printed epoxy, respectively. Laminate the prepared magnet-loaded PET cantilever onto a supporting epoxy ring, followed by another one layer of PET with prefabricated breathing holes.

### Fabrication of olfaction interfaces

Flexible circuits were developed by printed circuit board processing techniques based on copper (Cu, thickness 10 μm) plated with gold (thickness 50 nm) as the conductive layer. Additionally, a copper layer was applied to both sides by exposing circle electrodes and square patches. Low-temperature soldering paste (LF999, KELL YSHUN Technology Co. LTD) was used to bond and electrically connect all of the components, including the Bluetooth (WH-BLE103), the microcontroller (ATmega328p-mu), capacitors (14–22 pF), resistors (0.8 MΩ), crystal oscillator (16 MHz), and OGs, to corresponding contact pads on the Cu/PI substrate. After the integration of all components, PDMS (crosslink: PDMS = 1:20, 145 kPa, 2 mm thick) were poured onto the device fixed in customized molds, followed by curing at 70 °C for 24 hrs. Then, we could obtain the two-channel and 32-channel OG-based olfaction interfaces.

### Odor landscape simulation

The spatial-temporal distribution of odorants within airflow is determined by the fluid dynamics of the ambient atmosphere (with the Navier-Stoke equation) and the motion of the odorant within it. The equation that governs the dynamics of the odorant concentration inside the airflow is the advection-diffusion equation, as advection refers to bulk motion, and diffusion refers to Brownian motion. The balance between these two processes is described by the Péclet number[42], which is the ratio between the rate of advection and the rate of diffusion. Specific to our application, we only consider the average odor concentration when diffusion dominates the flow dynamics in crosswind direction with slow laminar flows. We calculated the spatial-temporal plume generated by N OGs with the Gaussian dispersion plume model:

$$C(x,y,z,h) = \frac{NQ}{2\pi V \sigma_y \sigma_z} \exp\left(\frac{-y^2}{2\sigma_y^2}\right) \cdot \left(\exp\left(\frac{-(z-h)^2}{2\sigma_z^2}\right) + \exp\left(\frac{-(z+h)^2}{2\sigma_z^2}\right)\right) \quad (4)$$

where $C(x, y, z, h)$ is the average concentration of diffusion substance at a point $(x, y, z)$ in OG's coordinate system, $x$ and $V$ represent the downwind direction and the average wind/airflow speed, and y stands for the crosswind axis. As for the case of OS 2 with slow horizontal wind, the last term model diffusions along vertical axis z and h are the effective stack height of the odor source. The width of the plume is governed by the rate of turbulent diffusion and is expressed in terms of the dispersion coefficients in crosswind axes. According to the less-than-2m/s wind and moderate daylight conditions, we determined the atmospheric stability of Pasquil classes[28] and applied the lateral dispersion coefficient function as $\sigma_y = 16x(1 + 0.0001x)^{-0.5}$, $\sigma_z = 0.12x$

according to Briggs expression table[27]. The source strength Q of each odor (g/sec) is calculated as the OG works at full load (the total operation hour of each OG is shown in Fig. 3f and Supplementary Fig. 20). For odor plume with slow vertical convective wind in still ambiance (OS 1, OS 3, and OS 4), downwind axis coincides with the vertical axis. The slow laminar convective air flows dominate the vertical axis. h turns to zero, and the model becomes an asymmetric diffusion model with $\sigma_y = \sigma_z = 0.2x$. Convective air velocities are calculated as $\dot{x} = V_c = (P/x)^{1/3}$, where P stands for the heat power from OG. With spatial-temporal models of odor plumes, if a user's location and nose height $h_u$ Were known, we could get the shortest distance, $\hat{D_{O_pU}}$, from the odor plume to the user's nose and the respective required odor transporting time $T_{trans}$. $\hat{D_{O_pU}}$ could also be interpolated as the potential distance from odor dispersion to human use.

### Operation of MR system

As shown in Supplementary Fig. 38, the positioning devices (DWM1000, Qorvo, Inc.) are attached to different objects in reality that will appear in the virtual environment, to track the 3-dimensional coordinate of different objects (also serving as OSs). The location information is transmitted to the computer via Bluetooth mesh and the data link is imported into the custom Unity program for the mixed reality application. Inside the Unity program, the decoded real-world coordinate data of the selected objects O(x, y, z) will be used to assign the three-dimensional coordinate of the corresponding virtual objects inside Unity's virtual space. When the real-world object moves in reality, the positioning system will capture the changing coordinates and update the virtual space in real-time. User's coordinates can be monitored via two different methods, for the small working area scenario (10 m × 10 m area), the VR glasses (Oculus Quest 2, Meta Platforms, Inc.) embedded motion and coordinate tracking system will be in charge of the coordinates of the user as well as the camera location in the virtual space. When the intended working area is bigger than 10 m × 10 m, the same positioning system used for object coordinate tracking will be attached to VR glasses to track the user's three-dimensional coordinates, U(x, y, z). The orientation of the user's head $\overline{head}$, and the virtual camera will be captured and controlled by the three-dimensional angle data from an inertial measurement unit integrated with the custom positioning system. When the user face towards odor source, $||<\overline{head}, \overline{UO}>|| \leq 45°$, as positioning system updates user's velocity and coordinates, the exploration potential toward an odor plume $O_p$, is updated as $\widehat{D_{UO_p}} = \frac{\overrightarrow{V_u} \cdot \overrightarrow{UO_p}}{|\overrightarrow{UO_p}|} \cdot (T_{response} + T_{trans})$, where $\overrightarrow{V_u}$ stands for user velocity, $\overrightarrow{UO_p}$ is the vector from user towards the downwind axis of an odor source, and $T_{response}$ is the response time of OGs. When the potential distance from odor dispersion plumes to the human user is less than the exploration potential, $\widehat{D_{O_pU}} \leq \widehat{D_{UO_p}}$, the system would demand related OGs to work.

An independent identity number will be assigned to the different reality objects and corresponding OG units. When the computer program determines specific OG unit needs to start working and emit certain odor information. The command will also be sent from the Unity program to a custom Bluetooth dongle that connects with the computer by serial communication, and the Bluetooth dongle will send the command including calculated small-intensity information to the target OG unit, which will release odor according to command. With the cooperation of the positioning system, VR headset, OG unit, Unity program, and odor intensity calculation algorithm, the mixed reality application of the OG unit can be achieved.

### Mechanical stimulation

The FEA commercial software ABAQUS (Analysis User's Manual 2020) was used to calculate the deformation of the structure. The PET, magnet, Cu, Epoxy and PI layers were modeled by 2.89 million linear

hexahedral elements of type C3D8R. Electrode composed of Au and Cr were modeled by 0.31 million linear quadrilateral elements of type S4R. The minimal element size was 0.001 mm, which ensured the convergence and the accuracy of the simulation results. The elastic modulus ($E$) and Poisson's ratio ($v$) used in the analysis were $E_{PET} = 3.5$ GPa, $v_{PET} = 0.35$, $E_{magnet} = 160$ GPa, $v_{magnet} = 0.24$, $E_{Cu} = 131$ GPa, $v_{Cu} = 0.33$, $E_{Epoxy} = 3.35$ GPa, $v_{Epoxy} = 0.35$, $E_{PI} = 3.15$ GPa, $v_{PI} = 0.34$, $E_{Electrode} = 79$ GPa and $v_{Electrode} = 0.41$.

## Characteristics

The odorous chemicals used in Figs. 5, 6 are ethanol (1), pineapple (2), grape (3), osmanthus (4), rice (5), tobacco (6), gardenia (7), watermelon (8), vanilla (9), coffee milk (10), candy (11), coconut milk (12), coconut (13), milk (14), peach (15), figue (16), orange (17), green tea (18), caramel (19), durian (20), lemon (21), strawberry (22), morning (23), ginger (24), clary sage (25), rosemary (26), lavender (27), clove (28), mojito (29), cake (30), cream (31), and pancake (32).

## Data availability

The data that support the findings of this study are available from the corresponding authors upon request.

## Code availability

The codes that support the findings of this study are available as following: https://doi.org/10.5281/zenodo.11147287.

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

## Acknowledgements

This work was supported by the Research Grants Council of the Hong Kong Special Administrative Region (Grant Nos. 11211523, RFS2324-1S03), National Natural Science Foundation of China (Grant No. 62122002), City University of Hong Kong (Grant Nos. 9667221, 9680322, 9678274), in part by InnoHK Project on Project 2.2—AI-based 3D ultrasound imaging algorithm at Hong Kong Centre for Cerebro-Cardiovascular Health Engineering (COCHE), the National Natural Science Foundation of China (Grant Nos. U23A20111 and 12372160), 111 Center (Grant No. B18002), Japan Society for the Promotion of Science (Grant No. 22K21343), and State Administration for Market Regulation Science and Technology Plan Project (Grant No. 2023MK201).

## Author contributions

YM.L., W.P., C.Y., and S.J. contributed equally to this work. YM.L., T.S., and X.Y. conceived the ideas, and designed the experiments. YM.L., W.P., X.Y., S.J., T.Y., and Z.Z. wrote the manuscript. YM.L., W.P., C.Y., S.J., Z.C., X.H., H.C., W.L., Y.G., and Y.Z. performed experiments and analyzed the experimental data. Z.Z., W.S., J.N., and YH.L. performed structural designs, mechanical and thermal modeling. S.J. performed algorithm designs.

## Competing interests

The authors declare no competing interests.

## Etihics statement

All procedures during the olfaction system testing from human participants are approved by the City University of Hong Kong Research Committee (HU-STA-00000194). The informed consent of all participants was obtained prior to inclusion in this study.
