## [Peer Review File · Nature Communications]

Intelligent Wearable Olfactory Interface for Latency-Free Mixed Reality and Fast Olfactory EnhancementREVIEWER COMMENTS

Reviewer #1 (Remarks to the Author):

This manuscript is a study about an olfactory interface based on miniaturized odor generators. The authors fabricated the generators and interface and adapted to a mixed reality system. This research is well organized and described. However, the authors' contribution is weak. Although the performance of the odor generators is improved, the concept is already reported in their previous paper (Ref. 2). This is not a research with new idea but succeeding one. With the improved performance about response time and power consumption, it is suitable to other engineering journals than nature communication.

Additional comments about the manuscript are below:

1. About the breathing holes, it needs to analyze the effect of size and numbers.
2. The injected perfume has limited operation time as shown in Fig3f. How did the operation time define? Is there the standard for operation time? Also, please show the degradation of odor along with the operation time.
3. At the valve, the shape with cantilever is difficult to fully close the flat surface. How about the leakage of odor?
4. In the array of odor generators, the gap between the generators affects recognition or training. Is it optimized? Please discuss the gap.
5. The wearable odor interface is mounted on forearm. However, forearm is far from nose. Is there a reason to select forearm? It needs the adjustment for the mounting position.

Reviewer #2 (Remarks to the Author):

This paper reports the development of latency-free olfactory feedback technologies by combining hardwares, i.e. wearable electronics, soft materials, and miniaturized odour generating devices, with software of advanced AI algorithms. The demonstration of mixed reality by using such olfactory feedback system showcases the technical advantages and application potentials. I feel there are two kinds of novelty of this work that impress me a lot, which I think should be also very attractive the research community and general audience. The first novelty is the advanced technology for this high performance olfactory generating system, as there are quite a lot of research reported the gas sensors/electronic noses, but the development of feedback is quite an unexplored area. The second novelty is the AI enabled latency-free feedback with a 70 ms response time, which is an encouraging results to show the great possibility of combining wearable electronics with AI. So, I feel this work is very impressive and deserve the publication in Nature Communications. Here are some comments for the authors to address.

1. The AI algorithms enhanced the performance of the olfactory system. How about the application of such AI into the traditional odour generating systems, i.e. the equipment in a room? Please put comments on this point.
2. There are some breathing holes in the top polymer layer. How the authors define the size and the arrangement of the breathing holes? I mean is the current report parameters are optimized?
3. The authors show different formats of the devices, i.e. mount on body, necklace, or even integrated with dummy flowers. Will the response time are all controlled and optimized by algorithms?
4. In the discussion part of olfactory training/recovery, what is the clinical gold standard to evaluate the treatment? Please add more information/evidence from references to support the conclusion.
5. I recommend the authors either to delete or add more information in the comparison of the device performance in this work to previous shown in Fig. 1f. Here you only compare this with 1 previous work, which is not convincing.

6. The measurement of ethanol as sensing parameter make sense. While I would suggest the authors to perform another test as the odour releasing duration vs temperature. You should get some good correlation between the ppm released per second vs temperature.

7. I feel the human olfactory threshold should be different to different odours. Please add discussion/outlook to this point.

Reviewer #3 (Remarks to the Author):

This paper presents the development of olfaction feedback technologies to overcome the limitation of current devices including human perceivable delay in odor manipulation, bulky size, limited number of odor supplies, and high-power consumption.

Considering the recent intense interest in the VR/AR, this may be recommended for publication if the authors can successfully respond to the following comments,

1) Latency-free olfactory MR concept is an interesting topic. Latency makes various problems in the VR/AR application. Especially, the latency in visual, auditory, and tactile sensation is very important. However, considering the olfactory sensation mechanism by mass transfer, the delay may be natural cause mass transfer by diffusion and convection takes a certain time to arrive at nose. I wonder if the latency-free concept is necessary in olfactory sensation. Instead of zero-latency, I think there must be optimum latency to generate the realistic olfactory feeling. Some discussion on this needs to be provided.

2) To assess the response time of OGs in controlling odors, the authors measured the real-time variations of ethanol concentration with the operation time at a surrounding airflow rate of 2.76 m/s, during which we closed the breathing hole at 0.72 s (Fig. 2i). The ethanol concentration was increased slightly from 745 ppm to 789 ppm in a short time (70 ms), then starts dropping down. The concentration enhancement at 0.72 s is caused by the internal ethanol gas ejected from the OGs, induced by the upward motion of the magnet. As a result, the authors determined that the response time of the OGs could be reduced to 70 ms. Here, I wonder if the response time can be decided to be 70 ms cause the ethanol concentration is still changing in the several second range and the slight increase may not be detected by nose. The further discussion needs to be provided.

3) As shown in the figure 5(b), the olfaction interface was mounted onto human forearm. Because there is some distance between olfaction interface and nose, it may take some time to for the OG to arrive at the nose. The successful and efficient olfaction interface had better be located near to the nose. The reason why the olfaction interface is located on the arm, and the possibility for miniaturization to be mounted on face near to the nose will provide the readers with the readers.

4) The OG adopts a solenoid valve structure to majorly control the odor release, where the breathing holes at the top PET layer could be opened and closed by manipulating the electromagnetic field. The olfactory interface supports 32 different common odor types, including fruits, spices, drinks, bakeries, herbs, and others. To minimize the unpredictable artifacts on the olfaction interface on these realistic models, all wire connections are hidden behind the OGs mounting surfaces. I wonder how 32 different order was selected and if 32 order types can generate any arbitrary smells.

5) Continuing the previous comments, 32 order types may be different diffusion rate even at the same temperature. Was each OG calibrated for these kind of diverse diffusion rate depending on the materials?

6) The authors only consider the average odor concentration when diffusion dominates the flow dynamics. To further simplify the model, they neglected the turbulent airflow and evaluated the effect of low-speed laminar flows in the odor diffusion process. The odor concentration gradients are calculated using the Gaussian plume model, considering a working OG as a constant release source. Cause the olfaction interface was mounted onto the human forearm. The dynamic environment condition needs to be considered because the arms will be usually swinging when the users are walking.

7) Unlike visual, auditory, and tactile sensory channels, olfaction is a nonlinear chemical sense, making it difficult to develop a technically comprehensive olfaction feedback system in precisely controlling odors generation and delivery. This study directly starts the discussion of olfactory interface without discussing general various other haptic feedbacks for artificial feeling generation (Nat. Rev. Mater., 7, 841, (2022); Adv. Funct. Mater., 31, 2106546 (2021)) including mechano-

haptic (mechanical artificial feeling, *Nature*, 575, 473 (2019); *Adv. Funct. Mater.*, 31, 2007772, (2021)), thermo-haptic (thermal artificial feeling, *Adv. Funct. Mater.*, 30, 1909171, (2020); *J. Mater. Chem. A*, 8, 8281 (2020); *Adv. Funct. Mater.*, 31, 2007376, (2021)). They need to be briefly discussed in the introduction part to provide the readers with the related recent progresses.

8) For the accurate OG control, the released chemical should be monitored and the active feedback to the controllers. I wonder if current system uses active control system or passive control system to accurately control the released amount of OG.

9) Some of the digital figures are missing scale bars. They should be added to the pictures.

10) The general usage of language including the typos and grammatical errors need to be checked again.

Responses to comments of Referee #1

Comments from Referee #1:

Summary Comment: This manuscript is a study about an olfactory interface based on miniaturized odor generators. The authors fabricated the generators and interface and adapted to a mixed reality system. This research is well organized and described. However, the authors' contribution is weak. Although the performance of the odor generators is improved, the concept is already reported in their previous paper (Ref2). This is not a research with new idea but succeeding one. With the improved performance about response time and power consumption, it is suitable to other engineering journals than nature communication.

Our response: We thank the referee for reviewing our work. This work is not only about the improvement of hardware, but also the combination with advanced algorithms to realize a nearly time delay free odor feedback system. We have made extensive revision, which we believe can highlight the novelty of this work.

Modifications: None

Comment 1: About the breathing holes, it needs to analyze the effect of size and number.

Our response: We thank the referee for this comment. In revision, we have conducted a series of experiments to assess the effect of breathing hole pattern on the performance of the OGs as shown in Supplementary Fig. SX1. We designed five different breathing hole patterns with different holes number and size (Supplementary Fig. 5b), and the solid area ratios of the five breathing hole patterns are calculated in Supplementary Fig. 5c. To investigate the breathing hole pattern effect, we conducted two experiments: one is the ethanol concentration variation as a function of the five breathing hole patterns (Supplementary Fig.5d, e), where the ethanol concentration generated by the OGs could be monitored by a commercial ethanol sensor (TGS823, Figaro Engineering Inc.); Another experiment is the user test to obtain the odor intensity as a function of the five different breathing hole patterns for 32 different odor types (see odor type details in Characteristics) in Supplementary Fig. 5f, g. Both of the two experiments have proven that higher breathing hole number and larger hole size could contribute to higher odor concentration, therefore, we select the optimized breathing hole pattern (Supplementary Fig. 5b) for the device. We have modified the content and figures for a clear illustration on Page 6 Line 15.

Modification: On Page 6 Line 15, we have modified the text as following “**Fig. 2** shows the result of a series of experimental optimizations performed on the breathing hole patterns, the internal heating temperature, the frequency, amplitude and duty cycle of the signal input into the electromagnetic coil, from which we can find that the optimized parameters enable the OG to exhibit ultra-fast response time, precise odor generation control, and low-power consumption. **Supplementary Fig. 5** presents the effect of breathing hole patterns in the top PET layer on the performance of the OGs, where five OGs adopt five different breathing hole patterns with different solid area ratios (0.914 ~ 0.996) of the PET layer (**Supplementary Figs. 5a, b, c**). **Supplementary Figs. 5d, e** show the ethanol concentration generated by the five OGs as a function of breathing hole patterns, and it is obvious that the OG#1 with the

lowest solid area ratio (0.914) could contribute to the highest stabilized ethanol concentration (~2649 ppm). To further investigate the effect of the breathing hole patterns on the electrical property of the OGs, we recorded the volunteers' sensed odor intensities to 32 different odor types (see details in Characteristics) generated by the five OGs (**Supplementary Figs. 5f, g**), where there are 6 males and 4 females in the volunteer test. It is found that the average odor intensity scores of the OG#1 is the highest among the five OGs, consistent with the results shown in **Supplementary Figs. 5d, e**. Therefore, the first breathing hole pattern (**Supplementary Fig. 5b**) is adopted."

Fig. S5. Optimization of the breathing hole pattern in the top PET layer of OGs. **a** Optical image of five OGs with different breathing hole patterns with enlarged details shown in **b**. **b** Designs of five different breathing hole patterns in the top PET layer of OGs. **c** Solid area ratios of the top PET layer as a function of the five different breathing hole patterns. **d** Ethanol concentration generated by the five OGs with different breathing hole patterns as a function of operation time. **e** The stabilized ethanol concentration generated by OGs as a function of the breathing hole patterns. Here, it is obvious that the OG#1 with a lowest solid area ratio of the top PET layer could contribute to a highest ethanol concentration (~2649 ppm). **f, g** A volunteer test showing the odor intensity sensed by 10 volunteers as a function of the five different breathing hole patterns for 32 different odor types (same to those in Fig. 3f). Here, there are 6 males and 4 females for the volunteer test. The odor intensity scores range from 0 to 3, corresponding to four odor intensities, including none, low, middle, and high. During the volunteer test, volunteers were asked what intensity score they sensed after smelling the same odors generated by the five OGs.

Comment 2: The injected perfume has limited operation time as shown in Fig. 3f. How did the operation time define? Is there the standard for operation time? Also, please show the degradation of odor along with the operation time.

Our response: We thank the referee for this comment. The operation time of the OG is defined as the duration when the OG starts to continuously release odor until that the odor intensity above the breathing holes of the OG is lower than the corresponding human olfactory perception threshold value. Therefore, to obtain the operation time shown in Fig. 3f, we conducted a volunteer test to collect the operation time of the 32 different odor types at a full-load operation mode. During the test, all volunteers are required to smell the odors generated by the OGs until they can not sense the odors. Here, volunteers were required to smell the odors every 5 minutes at least, and a higher smelling frequency is allowed. To investigate the degradation of odor along with the operation time, we conduct an experiment to monitor the ethanol concentration as a function of operation at three different heating temperatures, including 55°C, 45°C, and 35°C, as shown in Supplementary Fig. 6. It is found that higher heating temperature will result in a lower operation time, where 80 ppm is the human olfactory threshold value to ethanol (*Spacecraft Maximum Allowable Concentrations for Selected Airborne Contaminants: Volume 5 (Vol. 5). National Academies Press.2008*). For the adopted perfume based odorous chemicals, as the corresponding commercial smell sensors are not available in the market, we couldn't monitor the degradation of perfume-based odors along with the operation time. Therefore, according to the definition mentioned above, for the ethanol gas, we define the operation time as the duration when OG starts to continuously release odor until that the monitored odor concentration above the breathing holes of the OG is lower than 80 ppm. And for the perfume-based odors, we define the operation time as the duration when OG starts to continuously release odor until that volunteers can not sense the odor above the breathing holes of the OG. We have modified the content and figures for a clear illustration on Page 7 Line 11.

Modification: On Page 7 Line 11, we have modified the text as following “**Fig. 2c** shows the positive linear relationship between the peak ethanol concentration (odor release concentration) and the OG heating temperature, increasing from 35°C @ 2102 ppm to 55°C @ 2705 ppm. **Supplementary Fig. 6** presents the operation time of the three OGs with the constant 0.04 mL ethanol embedded as a function of the heating temperature ranging from 55°C to 35°C, where 80 ppm is the human olfactory threshold value to ethanol²³. Here, the operation time of the OG is defined as the duration when the OG starts to continuously release odor until that the odor intensity above the breathing holes of the OG is lower than the corresponding human olfactory perception threshold value. It is obvious that the higher heating temperature could result in a shorter operation time (**Supplementary Fig. 6c**), which may result from the more intense Brownian motion and air convection induced by larger temperature differences between the heating electrode and ambient air temperature. Considering the available commercial ethanol sensor (TGS823, Figaro Engineering Inc.) to detect the odor concertation, we chose the ethanol solution (vol%, 95%) as the odorous chemical stored inside OGs instead of other biosafe odorous chemicals.”

Fig. S6. a, c Ethanol concentration generated by the OGs as a function of operation time when the heating temperatures **(b)** of the OGs are targeted at 55°C, 45°C, and 35°C, respectively.

The newly added reference:

[23] Council, N. R. et al. *Spacecraft Maximum Allowable Concentrations for Selected Airborne Contaminants: Volume 5. Vol. 5* (National Academies Press, 2008).

Comment 3: At the valve, the shape with cantilever is difficult to fully close the flat surface. How about the leakage of odor?

Our response: We thank the referee for this comment. As shown in Fig. 1a, the surface roughness of the PET cantilever and the top PET layer with breathing holes has an influence on the air tightness of the OG. To investigate the odor leakage from the OG, we monitored the ethanol concentration generated by the OG with the heating temperature varying from 35°C to 55°C when the OG is programmed to continuously close the breathing holes (Supplementary Fig. 10). It is found that the leaked ethanol concentration at the outlet and 1.5 cm above OG could reach up to 255 ppm and 62 ppm, respectively. As 255 ppm is slightly lower than human minimum identifiable odor level (350 ppm) (*American Industrial Hygiene Association Journal* 19, 494-498, 1958), indicating that although the leaked alcohol concentration can be perceived by humans (human ethanol threshold value, 80 ppm), it cannot be identified by humans due to the extremely low concentration. Therefore, it is a fact that our OG will leak a small amount of odor with a low concentration. To avoid smelling the leaked odor, users are recommended to experience the odors generated by the olfactory interface at a distance between OGs and users' nose over 1.5 cm. We have modified the content and figures for a clear illustration on Page 10 Line 13.

Modification: On Page 10 Line 13, we have modified the text as following “To investigate the gas tightness of the OG, we monitored the ethanol concentration around the OG with the breathing holes closed, during which the heating temperature of the OG is programmed to increase from 35°C to 55°C for assessing the heating temperature effect on the gas leakage from the OG (**Supplementary Fig. 10**). It is found that the gas leakage is observed with a maximum ethanol concentration reaching up to 255 ppm at the heating temperature and distance between the OG and ethanol sensor of 55°C and 0 cm, respectively. The leaked ethanol concentration (255 ppm) is slightly lower than human minimum identifiable odor level (350 ppm)²⁴, indicating that although the leaked alcohol concentration can be perceived by humans, it cannot be identified by humans due to the extremely low concentration. When the distance between the OG and sensor is set at 1.5 cm, the leaked ethanol concentration ranges from 49 ppm to 62 ppm, which is out of human perception range, where the ethanol concentration at the distance of 1.5 cm during the odor releasing status could reach up to 1491 ppm (**Fig. 2I**). As a result, users are recommended to experience the odors generated by the

olfactory interface at a distance between OGs and users' nose over 1.5 cm to avoid smelling a small amount of leaked odors.”

Fig. S10. Ethanol concentration generated by the OG as a function of the distance between the ethanol sensor and the breathing holes of the OG, where the OG is programmed to continuously close the breathing holes. During the test, the heating temperature of the OG is generally increased from 35°C to 55°C to investigate the heating temperature effect on the leakage of ethanol gas from the OG.

The newly added reference:

[24] Scherberger, R., Happ, G. P., Miller, F. A. & Fassett, D. W. A dynamic apparatus for preparing air-vapor mixtures of known concentrations. *American Industrial Hygiene Association Journal* 19, 494-498 (1958).

Comment 4: In the array of odor generators, the gap between the generators affects recognition or training. Is it optimized? Please discuss the gap.

Our response: We thank the referee for this comment. The gap between the OGs has been optimized before we designed the 32-channel olfactory interface for realizing an optimal OGs array layout. We are sorry for the omissions of the optimization process. In the revised manuscript, we presented the optimization in Supplementary Figs. 21, 22. During the optimization experiment, we observed if the magnets motion inside OGs are affected when the two OGs are approaching to each other at the two different OG layouts, including side-by-side and corner-to-corner methods (Supplementary Figs. 22a, b, c), where the two OGs are programmed to operate at a constant frequency (0.5 Hz) of AC power into the electromagnetic coil. During the OGs approaching process, once the magnet movement cannot operate as normal, the distance between the OGs will be recorded as the shortest value. As a result, we found that once the shortest distance between the magnets center of the two OGs is over 1.5 cm, the magnet motion inside OGs will not be affected. Guided by the gap optimization, the 32-channel olfactory interface adopts the corner-to-corner OG layout method for achieving a higher OGs array density (0.75 OGs/cm²), compared to the that (0.55 OGs/cm²) of the side-by-side layout design (Supplementary Figs. 21, 22). We have modified the content and figures for a clear illustration on Page 15 Line 23.

Modification: On Page 15 Line 23, we have modified the text as following “Based these foundation and remaining challenges, in this new demonstration, a flexible, forearm-mounted olfaction interface with 32 OGs embedded is introduced, paired with a self-developed, flexible control panel connected by four bendable flat cables (Figs. 5a, b). Compared to the

face mask based olfactory interface with 12 OGs integrated, the forearm-mounted olfactory interface supplies more odor options in a more user-friendly method for a higher efficiency in conducting the olfactory training (**Fig. 5b and Supplementary Fig. 21**). Here, the gap between OGs is optimized by investigating the permanent magnetic force effect on the motion of the magnets inside the OGs (**Supplementary Fig. 22**), and it is found that the magnets of OGs could operate as normal once the distance between the center of the magnets is over 1.5 cm. In addition, the corner-to-corner OG layout (**Supplementary Fig. 22c**) could contribute to a higher OGs array density (0.75 units/cm^2) than that (0.55 units/cm^2) of the side-by-side layout (**Supplementary Figs. 22a, b**), as observed in **Figs. 5a, b and Supplementary Fig. 21**. Therefore, the 32-channel olfactory interface adopts the corner-to-corner OGs layout with a height gap of 2 mm (**Supplementary Figs. 22c, d, e**.)”

Fig. S21. a, b Optical images of the 4×3 OGs array mounted onto a self-designed FPCB with the top and bottom views. c Optical image of the face-mask olfactory interface based on the 4×3 OGs array.

Fig. S22. a, b, c Optical images of the three two-OG layouts. d, e Optical images of the 32-channel olfaction interface with enlarged details for presenting the OGs layout method.

Comment 5: The wearable odor interface is mounted on forearm. However, forearm is far from nose. Is there a reason to select forearm? It needs the adjustment for the mounting position.

Our response: We thank the referee for this comment. We agree that the distance between the wearable odor interface and the nose highly affects the performance of the interface, and the mounting of the device on the forearm is more distal compared to the skin region near the mouth. We adopt the forearm as the application area for the wearable olfactory interface by considering the number of integrated OGs, paired power sources, user friendliness, and convenient odorous chemicals refilling.

Number of integrated OGs. The wearable 32-channel olfactory interface is developed to realize the efficient olfactory recovery, therefore, the most critical issue is how to integrate as many OGs as possible in a limited mounted skin area instead of the response time required to transmit the generated odors to human nose as more supplied odors could increase the improvement rate of olfactory function (*Journal of Clinical Medicine* 12, 4761 (2023)). The forearm has a larger covering space for OG array assembly than the skin region near the mouth, and it is a body part that can easily approach the nose while a user can freely move and adjust displacement voluntarily. By mounting the olfactory interface onto the forearm, the intensity of odors can be determined based on a user's preference by adjusting the distance between the device and user's nose.

Paired power sources. The 32-channel olfactory interface requires a high-power consumption, which could be supplied by the commercial Li-ion battery (10 V, 1500 mAh) or an external DC power source. By mounting the olfactory interface onto the forearm, users could flexibly choose the two types of the power sources according to their preferences in different practical applications.

User friendliness. Here, we developed a mouth-based system by exploiting a facemask to compare the operational efficiency of the forearm and mouth-based interfaces (Fig. 5b, and Supplementary Fig. 21). As a result, distinct drawbacks of the facial mask olfaction interface were observed, especially in the aspect of user-friendliness. First of all, due to the structure of the mask completely covering the mouth and nose, the released odors were not able to be hastily diffused out, and inconvenience to users was induced during respiration, especially during the silence of the olfaction interface. In addition, as shown in Supplementary Fig. 26, the large-area control panel and power sources will require additional skin areas around users' face, furtherly affecting users' head movement.

Odorous chemicals refilling. Lastly, compared to the forearm-based olfaction interface, the difficulty in refilling odorous chemicals inside OGs increase a lot due to the limitation in wearable formats. As shown in Supplementary Fig. 21c the olfaction interface is entirely embedded in the facial mask, and therefore, to refill the odorous chemicals, the mask must be taken off from the user. Instead, the odorous chemicals can be easily injected into OG units for the forearm-based olfaction interface, even during operation. As a result, we adopted the forearm to mount the olfactory interface. We have modified the content and figures for a clear illustration on Page 15 Line 16.

Modification: On Page 15 Line 16, we modified the text as following "It is believed that olfactory training can induce brain neural remodeling and increase the volume of the olfactory bulb, therefore significantly improving the ability of olfactory recognition, discrimination, and the olfactory perception valve³⁰. In addition, some studies have found that increasing the types of odors can increase the improvement rate of olfactory function³¹. Therefore, considering the number of integrated OGs, paired power sources, user friendliness, and convenient odorous chemicals refilling, the forearm is selected as the mounting position due to the large skin surface area and high accessibility to human nose. Based these foundation and remaining

challenges, in this new demonstration, a flexible, forearm-mounted olfaction interface with 32 OGs embedded is introduced, paired with a self-developed, flexible control panel connected by four bendable flat cables (**Figs. 5a, b**). Compared to the face mask based olfactory interface with 12 OGs integrated, the forearm-mounted olfactory interface supplies more odor options in a more user-friendly method for a higher efficiency in conducting the olfactory training (**Fig. 5b and Supplementary Fig. 21**). Here, the gap between OGs is optimized by investigating the permanent magnetic force effect on the motion of the magnets inside the OGs (**Supplementary Fig. 22**), and it is found that the magnets of OGs could operate as normal once the distance between the center of the magnets is over 1.5 cm. In addition, the corner-to-corner OG layout (**Supplementary Fig. 22c**) could contribute to a higher OGs array density (0.75 units/cm^2) than that (0.55 units/cm^2) of the side-by-side layout (**Supplementary Figs. 22a, b**), as observed in **Figs. 5a, b and Supplementary Fig. 21**. Therefore, the 32-channel olfactory interface adopts the corner-to-corner OGs layout with a height gap of 2 mm (**Supplementary Figs. 22c, d, e**).”.

Fig. S21. a, b Optical images of the 4×3 OGs array mounted onto a self-designed FPCB with the top and bottom views. c Optical image of the face-mask olfactory interface based on the 4×3 OGs array.

Responses to comments of Referee #2

Comments from Referee #2:

Summary Comment: This paper reports the development of latency-free olfactory feedback technologies by combining hardware, i.e. wearable electronics, soft materials, and miniaturized odour generating devices, with software of advanced AI algorithms. The demonstration of mixed reality by using such olfactory feedback system showcases the technical advantages and application potentials. I feel there are two kinds of novelty of this work that impress me a lot, which I think should be also very attractive to the research community and general audience. The first novelty is the advanced technology for this high performance olfactory generating system, as there are quite a lot of research reported the gas sensors/electronic noses, but the development of feedback is quite an unexplored area. The second novelty is the AI enabled latency-free feedback with a 70 ms response time, which is an encouraging result to show the great possibility of combining wearable electronics with AI. So, I feel this work is very impressive and deserves the publication in Nature Communications. Here are some comments for the authors to address:

Our response: We thank the referee for the positive comment.

Modification: None.

Comment 1: The AI algorithms enhanced the performance of the olfactory system. How about the application of such AI into the traditional odour generating systems, i.e. the equipment in a room? Please put comments on this point.

Our response: We thank the referee for this comment. We think that the performance of the conventional odor-generating systems could be enhanced significantly with the combination of reporting AI algorithms for realizing latency-free MR and fast olfactory training. However, there are some shortcomings in utilizing the conventional odor generating system with AI integrated as following:

First of all, the size and response time of the equipment are immensely important for adopting the latency-free algorithm in MR. The reporting olfaction device is much smaller with a faster response time compared to the traditional olfaction systems. In MR applications, the latency-free algorithm is triggered based on the distance between a user and odor sources based on the odor generating devices (or equipment) in MR. Therefore, if the odor sources based on the big equipment are placed very near, the algorithm for latency-free olfactory feedback in MR will always trigger the operation of the big equipment due to the second-level response time, which has been proven in Fig. 4. In addition, in the MR demonstration, we tried to integrate the OGs with the dummy flowers, fruits, food, and drinks to achieving a realistic tactile perception for users when they touch the virtual objects in the virtual environment. However, it is difficult to integrate the bulky equipment with these small objects. Furthermore, unlike the reporting olfaction device, the conventional ones are inconvenient for relocation to match the designed MR environment on account of their heavy weight and bulkiness, and a large space is also required to place the equipment. Similarly, the size and weight of the conventional odor-generating systems incur a mismatch with the fast olfactory training algorithm. The algorithm is mainly developed to enable a user to process training

themselves anywhere promptly, and therefore, the olfaction interface is required to be wearable. As a result, if the conventional equipment is too bulky to carry around, users have to visit a specific location for olfactory training (like hospital), which is not user-friendly. We have modified the content and figures for a clear illustration on Page 3 Line 23.

Modification: On Page 3 Line 23, we have modified the text as following “Apart from the challenges in latency issue of OGs, a number of technical difficulties still exist, including miniaturizing the size, optimizing the power consumption, and building up high channel counts/density¹⁸. For instance, the state of art odors generating methods generally require a whole set of bulky equipment with limited odors supplied and professional operation, forcing users to visit a specific place for experiencing olfactory feedback every time, which is unfriendly to users¹⁹.”

On Page 15 Line 6, we have modified the text as following “By demanding OGs to work at the temporal-spatial crossing point of these two potentials, the algorithm achieves latency-free experiences with minimal electrical energy waste. To illustrate the effect of response time, we also simulated the exploration and control process with a 1.4-s response time from the previously reported OG². In contrast to **Figs. 4h, i, j**, the operation time of the new OG integrated in OS 2 is 4.1 s, approximately one quarter of that of the previous OG (15.1 s). Therefore, considering the second-level response, bulky size, and high-power consumption, the conventional odor generation equipment is not suitable for the latency-free MR applications.”

Comment 2: There are some breathing holes in the top polymer layer. How the authors define the size and the arrangement of the breathing holes? I mean is the current report parameters are optimized?

Our response: We thank the referee for this comment. We didn’t take the optimization of the breathing hole pattern in the top PET layer into the consideration in the original manuscript as we think the heating temperature, the frequency, amplitude and duty cycle of the power input into the electromagnetic coil may be the efficient methods to control the odor release rate. To evaluate the impact of breathing hole patterns on the performance of the OGs, we designed the five different breathing hole patterns in terms of hole numbers and sizes (Supplementary Figs. 5a, b), where the solid area ratios of the five breathing hole patterns range from 0.914 to 0.996 (Supplementary Fig. 5c). Here, we designed two experiments: one is the ethanol concentration variation as a function of the five breathing hole patterns (Supplementary Figs. 5d, e), where the ethanol concentration generated by the OGs could be monitored by a commercial ethanol sensor (TGS823, Figaro Engineering Inc.); Another one experiment is an volunteer test to obtain the odor intensity sensed by 10 volunteers as a function of the five different breathing hole patterns for 32 different odor types (see odor type details in Characteristics) in Supplementary Figs. 5f, g. Both of the two experiments have proven that higher breathing hole number and larger hole size could contribute to higher odor concentration, therefore, we select the first breathing hole pattern (Supplementary Fig. 5b) as the optimal one. We have modified the content and figures for a clear illustration on Page 6 Line 15.

Modification: On Page 6 Line 15, we have modified the text as following “**Fig. 2** shows the result of a series of experimental optimizations performed on the breathing hole patterns, the

internal heating temperature, the frequency, amplitude and duty cycle of the power input into the electromagnetic coil, from which we can find that the optimized parameters enable the OG to exhibit ultra-fast response time, precise odor generation control, and low-power consumption. **Supplementary Fig. 5** presents the effect of breathing hole patterns in the top PET layer on the performance of the OGs, where five OGs adopt five different breathing hole patterns with different solid area ratios (0.914 ~ 0.996) of the PET layer (**Supplementary Figs. 5a, b, c**). **Supplementary Figs. 5d, e** show the ethanol concentration generated by the five OGs as a function of breathing hole patterns, and it is obvious that the OG#1 with the lowest solid area ratio (0.914) could contribute to the highest stabilized ethanol concentration (~2649 ppm). To further investigate the effect of the breathing hole patterns on the electrical property of the OGs, we recorded the volunteers' sensed odor intensities to 32 different odor types (see details in Characteristics) generated by the five OGs (**Supplementary Figs. 5f, g**), where there are 6 males and 4 females in the volunteer test. It is found that the average odor intensity scores of the OG#1 is the highest among the five OGs, consistent with the results shown in **Supplementary Figs. 5d, e**. Therefore, the first breathing hole pattern (**Supplementary Fig. 5b**) is adopted."

Fig. S5. Optimization of the breathing hole pattern in the top PET layer of OGs. **a** Optical image of five OGs with different breathing hole patterns with enlarged details shown in **b**. **b** Designs of five different breathing hole patterns in the top PET layer of OGs. **c** Solid area ratios of the top PET layer as a function of the five different breathing hole patterns. **d** Ethanol concentration generated by the five OGs with different breathing hole patterns as a function of operation time. **e** The stabilized ethanol concentration generated by OGs as a function of the breathing hole patterns. Here, it is obvious that the OG#1 with a lowest solid

area ratio of the top PET layer could contribute to a highest ethanol concentration (~2649 ppm). **f, g** A volunteer test showing the odor intensity sensed by 10 volunteers as a function of the five different breathing hole patterns for 32 different odor types (same to those in Fig. 3f). Here, there are 6 males and 4 females for the volunteer test. The odor intensity scores range from 0 to 3, corresponding to four odor intensities, including none, low, middle, and high. During the volunteer test, volunteers were asked what intensity score they sensed after smelling the same odors generated by the five OGs.

Comment 3: The authors show different formats of the devices. i.e. mount on body, necklace, or even integrated with dummy flowers. Will the response time are all controlled and optimized by algorithms?

Our response: We thank the referee for this comment. The response time of the olfactory interfaces directly mounted onto human body cannot be controlled by algorithms but is a fixed response time of 70 ms (Fig. 2i) as the latency-free algorithm requires inputs of various parameters, including the 3D position of the device, ambient temperature, wind speed, and air pressure (Figs. 3a, g). These parameters need bulky sensors for data collection, which bring great difficulties to the integration of all sensors with the skin-mounted devices. However, if the olfactory interfaces are not mounted onto human body, any olfactory interfaces could realize a controllable response time as it is easy to integrate the sensors with the olfactory interfaces (Fig. 4 and Movie S3). As a result, as observed in Fig. 3b and Supplementary Figs. 12, 13), we could develop various formats of the olfactory interfaces, ranging from dummy flowers, to dummy food, to dummy fruit, to necklace, to ring, and to drinks. As long as these interfaces are not mounted onto human body, the latency-free algorithm could be adopted to control their response time. We have modified the content and figures for a clear illustration on Page 13 Line 21.

Modification: On Page 13 Line 21, we have modified the text as following “The positioning system could be mounted onto different objects in reality to track their 3-dimensional coordinates in real time (**Fig. 3g**). Here, it is worth mentioning that due to the bulky size of the positioning system, it is difficult to integrate the system with skin-mounted olfactory interfaces, which means that the skin-mounted olfactory interfaces cannot achieve a latency-free MR experience. Following, the 3D location information is wirelessly transmitted to the computer via Bluetooth mesh. By constructing virtual objects with the same 3D coordinates as that in reality, we could link virtuality and reality to realize the MR experience for users.”

Comment 4: In the discussion part of olfactory training/recovery, what is the clinical gold standard to evaluate the treatment? Please add more information/evidence from references to support the conclusion.

Our response: We thank the referee for this comment, Despite the extensive history of research on olfactory perception, and a huge amount of olfactory evaluation methods has been employed. However, to the best of our knowledge, there is currently no widely adopted gold standard for olfactory evaluation within academia. Monitoring brain activity through MRI has been utilized in some studies for evaluation (Journal of the Neurological Sciences, 2012. 323(1): p. 16-24.), while olfactory testing remains the most prevalent method for assessing

olfactory dysfunction or recovery. Various studies may employ different tests for this purpose, including but not limited to: University of Pennsylvania Smell Identification Test (UPSIT)^{23, 24}, Connecticut Chemosensory Clinical Research Center Test (CCCRC)^{25,26}, Barcelona Smell Test (BAST-24)^{27,28}, Snap & Sniff Olfactory Test System^{29,30}, “Sniffin’ Sticks” Test^{31,32}, and more. Different test having their own method and standard in olfactory performance evaluation. For instance, the UPSIT test necessitates participants to evaluate 10 distinct odorants and subsequently respond to multiple-choice questions comprising four options, with their responses being scored accordingly for accuracy. Conversely, the CCCRC test presents participants with pairs of aqueous solutions differing in concentration, prompting them to discern the bottle containing the more potent scent; an escalated concentration is administered upon an incorrect response until the correct identification is achieved. In contrast, the BAST-24 test entails the presentation of a total of 24 olfactory stimuli, whereby participants are tasked with completing a questionnaire. This questionnaire probes participants on their ability to detect and recognize the presented odors, as well as to select the correct odor from a set of four options. However, in essence, these tests are all based on smell identification or threshold testing, similar to the methodology employed in our study, where we collected tens of volunteers’ recognition rates and reaction time to the 32 different odors (Figs. 5, 6). Consequently, we believe our testing approach is also suitable for olfactory evaluation. We have modified the content and figures for a clear illustration on Page 17 Line 16.

Modification: On Page 17 Line 16, we have modified the text as following “To demonstrate the high-channel olfaction interfaces, we called an experimental group with 11 volunteers (5 males and 6 females) and a control group with 14 volunteers (8 males and 6 females) to test human olfactory capability variations with and without 1-hr pretraining time, respectively, in terms of recognition rate and reaction time in distinguishing 32 different odors generated randomly by the olfaction interface (**Fig. 5c-g**). Here, various olfactory training tests have been designed, encompassing but not confined to: University of Pennsylvania Smell Identification Test (UPSIT)^{30,31}, Connecticut Chemosensory Clinical Research Center Test (CCCRC)^{32,33}, Barcelona Smell Test (BAST-24)^{34,35}, Snap & Sniff Olfactory Test System^{36,37}, and "Sniffin' Sticks" Test^{38,39}. Although these tests employ distinct protocols and evaluation criteria without a universal gold standard, they are rooted in evaluating users’ odor identification or threshold. Therefore, we collected volunteers’ recognition rates and reaction time to the 32 different odors for assessing their odor identification and threshold abilities. During the volunteer test, all volunteers were required to wear the 32-channel OGs based olfaction interface system on their forearms, then the experimenter manipulated the self-developed GUI installed in a personal computer for generating a random odor, where the tested volunteer only had one chance to give their answer without time limit for each odor.”

The newly added references:

[32] Doty, R. L., Shaman, P., Kimmelman, C. P. & Dann, M. S. *University of Pennsylvania Smell Identification Test: a rapid quantitative olfactory function test for the clinic. The Laryngoscope* 94, 176-178 (1984).

[33] Doty, R. L., Bromley, S. M. & Stern, M. B. *Olfactory Testing as an Aid in the Diagnosis of Parkinson's Disease: Development of Optimal Discrimination Criteria. Neurodegeneration* 4, 93-97 (1995). <https://doi.org/https://doi.org/10.1006/neur.1995.0011>

[34] Veyseller, B. et al. *Connecticut (CCCRC) olfactory test: normative values in 426 healthy volunteers. Indian Journal of Otolaryngology and Head & Neck Surgery* 66, 31-34 (2014).

- [35] Cain, W. S., Goodspeed, R. B., Gent, J. F. & Leonard, G. Evaluation of olfactory dysfunction in the Connecticut chemosensory clinical research center. *The Laryngoscope* 98, 83-88 (1988).
- [36] Cardesín, A. et al. Barcelona Smell Test-24 (BAST-24): validation and smell characteristics in the healthy Spanish population. *Rhinology* 44, 83 (2006).
- [37] Langdon, C. et al. Olfactory training in post-traumatic smell impairment: mild improvement in threshold performances: results from a randomized controlled trial. *Journal of neurotrauma* 35, 2641-2652 (2018).
- [38] Doty, R. L. et al. in *International forum of allergy & rhinology*. 986-992 (Wiley Online Library).
- [39] Jiang, R.-S. & Liang, K.-L. A pilot study of the snap & sniff threshold test. *Annals of Otolaryngology, Rhinology & Laryngology* 127, 312-316 (2018).
- [40] Kobal, G. et al. "Sniffin'sticks": screening of olfactory performance. *Rhinology* 34, 222-226 (1996).
- [41] Orhan, K. S., Karabulut, B., Keleş, N. & Değer, K. Evaluation of factors concerning the olfaction using the Sniffin' Sticks test. *Otolaryngology--Head and Neck Surgery* 146, 240-246 (2012).

Comment 5: I recommend the authors either to delete or add more information in the comparison of the device performance in this work to previous shown in Fig. 1f. Here you only compare this with 1 previous work, which is not convincing.

Our response: We thank the referee for this comment. As Fig. 1f could assist readers to accurately find the scientific novelty of the paper in a short time, we hope to keep this figure by adding one more published work for comparison. Here, there are two reasons for the limited research works for the comparison: (1) there are extremely limited research works reporting the wearable olfactory interface systems; (2) Among all the relevant research works, very few works could provide detailed electrical and mechanical properties of their olfactory interfaces, as shown in Supplementary Table 1. We have modified the content and figures for a clear illustration on Page 6 Line 7.

Modification: On Page 6 Line 7, we have modified the text as following “The core technical parameters of the OG reported in this work show significant improvement compare to the state of art ones, including the response time in controlling odor release, the lowest operating power consumption, overall size, stability of repeatedly switching the heating temperatures, minimum operation temperature, and maximum number of available odor types in one olfaction interface (**Fig. 1f**)^{2,4}. Due to the excellent performances of the new OG, the olfaction interfaces based on it can be utilized in various scenarios ranging from MR based entertainment, education, olfaction recovering, and enhancement (**Fig. 1g**).”

Fig. 1. Concept of the newly developed olfaction interfaces. a) Exploded view of the odor generator with detailed descriptions on each component. b) Optical images and schematic diagram of the OG with the additional inner views. c) Manipulation methods of the OGs in controlling odor generation rate. d, e) Mechanical stimulations of the Au based heating electrode (d) and cantilever-like PET film (e) under the distributed external pressure and bending deformation, respectively. f) Comparison of the new OGs and the recently reported ones in terms of response time, lowest power consumption in operating OGs, overall size, stability, operation temperature, and the maximum number of supply odors. g) Schematic diagram of the advanced OGs based olfaction interfaces for providing olfaction feedbacks to users at four typical applications, ranging from MR based entertainment and education to olfaction training for olfactory recovery and enhancement.

Table S1. Comparison of our work with the olfactory interfaces reported recently.

Working principle	Response time (s)	Size, L × W × H (mm ³)	Number of supply odors	Power consumption (mW)	Applications	Reference
Piezoelectric atomizer	N/A	Diameter of 14 mm, and length of 56.8 mm for the container	1	N/A	Influence emotions and cognitive performance	3
Vaporization odorous chemicals by heating, then delivered by fan	A few seconds	84 × 59 × 31	8	N/A	Influencing emotions, and alerting upcoming event	14
Physical phase change of odorous paraffin wax	30	130 × 110 × 80	6	N/A	Olfaction training	15
Atomizer	6	110 × 56 × 35	1	N/A	VR game	16
Blowing odor vapor by fans	> 1	527 × 60 × 328	1	N/A	VR game	24
Atomizer	5.16	N/A	4	N/A	VR game	25
SAW atomizer	6	Length, over 20 cm	8	N/A	Multimedia content and VR	26
Physical phase change of odorous paraffin wax	1.4 s	18 × 16 × 3 for each OG	9	220.4	VR, 4D movie, smell message delivery, clinical treatment, and emotion smoothing.	Our previous work, 2
Physical phase change of odorous paraffin wax	1 s	11 × 10 × 2.2 for each OG	8	317.4	Message delivery	Our previous work, 4
Solenoid valve	0.07 s	11 × 10 × 1.8 for each OG	32	84.8	MR and olfaction training.	This work

Comment 6: The measurement of ethanol as sensing parameter make sense. While I would suggest the authors to perform another test as the odor releasing duration vs temperature. You should get some good correlation between the ppm released per second vs temperature.

Our response: We thank the referee for this comment. We have made up a new experiment to investigate the relationship between the operation time and heating temperature of OGs, as shown in Supplementary Fig. 6. During the test, the three OGs with the 0.04 ml ethanol embedded were programmed to continuously release ethanol gas at a 20% duty cycle of the power input into the electromagnetic coil, where the heating temperatures of the three OGs were set at 55°C, 45°C, and 35°C, respectively. Then, we adopted a commercial ethanol sensor (TGS823, Figaro Engineering Inc.) to monitor the generated ethanol concentration variations for obtaining the operation time. Here, the end time of the ethanol release process is the time when the ethanol concentrations drop to 80 ppm (human olfactory threshold value to ethanol gas). As a result, we found that the higher heating temperature could result in a shorter operation time (Supplementary Fig. 6c), which may result from the more intense Brownian motion and air convection induced by larger temperature differences between the heating electrode and ambient air temperature. We have modified the content and figures for a clear illustration on Page 7 Line 11.

Modification: On Page 7 Line 11, we have modified the text as following “**Fig. 2c** shows the positive linear relationship between the peak ethanol concentration (odor release concentration) and the OG heating temperature, increasing from 35°C @ 2102 ppm to 55°C @ 2705 ppm. **Supplementary Fig. 6** presents the operation time of the three OGs with the constant 0.04 mL ethanol embedded as a function of the heating temperature ranging from 55°C to 35°C, where 80 ppm is the human olfactory threshold value to ethanol²³. Here, the operation time of the OG is defined as the duration when the OG starts to continuously release odor until that the odor intensity above the breathing holes of the OG is lower than the corresponding human olfactory perception threshold value. It is obvious that the higher heating temperature could result in a shorter operation time (**Supplementary Fig. 6c**), which may result from the more intense Brownian motion and air convection induced by larger temperature differences between the heating electrode and ambient air temperature. Considering the available commercial ethanol sensor (TGS823, Figaro Engineering Inc.) to detect the odor concentration, we chose the ethanol solution (vol%, 95%) as the odorous chemical stored inside OGs instead of other biosafe odorous chemicals.”

Fig. S6. a, c Ethanol concentration generated by the OGs as a function of operation time when the heating temperatures (**b**) of the OGs are targeted at 55°C, 45°C, and 35°C, respectively.

Comment 7: I feel the human olfactory threshold should be different to different odors. Please add discussion/outlook to this point.

Our response: We thank the referee for this comment. Actually, we are very interested to investigate the reviewer’s conjecture, and we also think it may be correct. To verify the

conjecture, we conducted a new volunteer test with 10 volunteers involved (6 males and 4 females) for assessing their olfactory thresholds to 32 different odors (see details in Characteristics) generated by the 32 OGs (Supplementary Figs. 11a-j). During the test, we recorded the volunteers' reaction time to the 32 different odors at a constant distance between volunteers' nose and the OGs of 1.5 cm. As seen in Fig. 2g, the longer reaction time could result in a higher odor concentration around the OGs, demonstrating a higher olfactory threshold value to the generated odor. As a result, as observed in Supplementary Figs. 11a-j, each volunteer exhibits different thresholds for the 32 different odors, and different volunteers also have distinguished thresholds for a same odor (Supplementary Fig. 11k). We have modified the content and figures for a clear illustration on Page 10 Line 21.

Modification: On Page 10 Line 21, we have modified the text as following “When the distance between the OG and sensor is set at 1.5 cm, the leaked ethanol concentration ranges from 49 ppm to 62 ppm, which is out of human perception range, where the ethanol concentration at the distance of 1.5 cm during the odor releasing status could reach up to 1491 ppm (**Fig. 2l**). As a result, users are recommended to experience the odors generated by the olfactory interface at a distance between OGs and users' nose over 1.5 cm to avoid smelling a small amount of leaked odors. **Supplementary Figs. 11a-j** present a volunteer test with 10 volunteers involved to investigate if human olfactory thresholds are distinguished for 32 different odors. During the test, we recorded the volunteers' reaction time when the volunteers were required to continuously smell the odors at a constant distance between the OGs and volunteers' nose until they could obviously sense the generated odors. It is interesting to find that the same volunteer may have different olfactory thresholds for different odors, and different volunteers also have distinguished thresholds for a same odor (**Supplementary Fig. 11k**).”

Fig. S11. A volunteer test showing that the odor reaction time of the 10 volunteers as a function of 32 different odor types (see odor details in Characteristics) at a constant distance between the OGs and the volunteers' nose of 1.5 cm. As observed in Fig. 2g, the longer reaction time to a specific odor could contribute to a higher odor concentration accumulated around the OGs, corresponding to a higher olfactory threshold value to the odor. Therefore, it is concluded that the same person may have different olfactory thresholds for different odors, and different people may also have distinguished thresholds for a same odor.

Responses to comments of Referee #3

Comments from Referee #3:

Summary Comment: This paper presents the development of olfaction feedback technologies to overcome the limitation of current devices including human perceivable delay in odor manipulation, bulky size, limited number of odor supplies, and high-power consumption. Considering the recent intense interest in the VR/AR, this may be recommended for publication if the authors can successfully respond to the following comments:

Our response: We thank the referee for the positive comment.

Modification: None.

Comment 1: Latency-free olfactory MR concept is an interesting topic. Latency makes various problems in the VR/AR application. Especially, the latency in visual, auditory, and tactile sensation is very important. However, considering the olfactory sensation mechanism by mass transfer, the delay may be natural cause mass transfer by diffusion and convection takes a certain time to arrive at nose. I wonder if the latency-free concept is necessary in olfactory sensation. Instead of zero-latency, I think there must be optimum latency to generate the realistic olfactory feeling. Some discussion on this needs to be provided.

Our response: We thank the referee for this inspiring comment. We agree that delayed olfactory sensation is natural in reality, but it is also necessary in the virtual environment. Once the olfaction feedback system has the latency-free property, we could program the device with the controllable delay time to mimic the real scenarios in the virtual environment. For example, when an individual quickly grabs a cup of coffee to their nose in the real world, the smell sensation can be generated nearly instantaneous. Conversely, when a person is standing near some flowers in reality, he will generally sense the fragrance of flowers after a certain delay, where there are many parameters in determining the delay time, including but not limited to ambient wind speed, air pressure, ambient air temperature, the diffusion rates of odorous chemicals, and the distance between the odor sources and human nose. The concept of zero-latency olfactory feedback encompasses the capacity to address various scenarios, including both immediate and delayed feedback. By setting a certain delay time in the AI system adopted for the latency-free MR applications, we could control the delay of olfactory feedback in the virtual environment, simulating the realistic human olfactory perception. In addition, latency-free olfactory feedback is crucial for scenarios necessitating instantaneous olfactory perception generation, an aspect not addressed by existing systems or research to the best of our knowledge. Therefore, we believe that a latency-free olfactory feedback system is important and necessary for better reproduction of real-world experiences. We have modified the content and figures for a clear illustration on Page 11 Line 10.

Modification: On Page 11 Line 10, we have modified the text as following “Under the umbrella of a self-developed AI algorithm, the olfactory interface based on the OGs could be utilized in MR for realizing latency-free applications. The latency-free olfactory system is not limited to only providing olfactory feedback with zero delay, but can achieve a controllable

olfactory feedback delay by setting a delay time in the AI algorithm, thereby simulating the real human olfactory perception in a virtual environment.”

Comment 2: To assess the response time of OGs in controlling odors, the authors measured the real-time variations of ethanol concentration with the operation time at a surrounding airflow rate of 2.76 m/s, during which we closed the breathing hole at 0.72 s (Fig. 2i). The ethanol concentration was increased slightly from 745 ppm to 789 ppm in a short time (70 ms), then starts dropping down. The concentration enhancement at 0.72 s is caused by the internal ethanol gas ejected from the OGs, induced by the upward motion of the magnet. As a result, the authors determined that the response time of the OGs could be reduced to 70 ms. Here, I wonder if the response time can be decided to be 70 ms cause the ethanol concentration is still changing in the several second range and the slight increase may not be detected by nose. The further discussion needs to be provided.

Our response: We thank the referee for this comment. The response time of the OG is defined as the delay time in generating or terminating odors, which is induced by the magnet movement inside the OGs. To investigate the response time of the OGs, we adopted a commercial ethanol sensor (TGS823, Figaro Engineering Inc.) to quantitatively monitor the ethanol concentration variations generated by the OGs, during which we started to close the breathing holes at the time point of 0.72s (Fig. 2i). As shown in Fig. 2i, a slight concentration enhancement from 0.72 s to 0.79 s can be observed, which is induced by the ejected ethanol gas from OGs. As the ethanol concentration is measured by the commercial sensor instead of volunteers, the slight ethanol concentration could be accurately detected, not by human nose. Here, it is worth mentioning that the high ambient airflow rate (2.76 m/s) could rapidly accelerate the ethanol diffusion, resulting in an obvious ethanol concentration drop after closing the breathing holes of the OGs, which has been proven in Fig. 2h. To furtherly verify the response time value of the OGs, we conducted a new experiment with the ambient airflow rate of 0 m/s, as shown in Supplementary Fig. 8, where the OG is programmed to close its breathing holes at the time point of 0.33 s. It is obvious that a slight ethanol concentration enhancement (from 718 to 720 ppm) could also be observed with a time gap of 70 ms, which is same as the value in Fig. 2i. As a result, we confirm that 70 ms is the response time of the new OGs. We have modified the content and figures for a clear illustration on Page 3 Line 14.

Modification: On Page 3 Line 14, we have modified the text as following “Up to now, the developed wearable olfactory interface systems are mainly built on arrays of odor generators (OGs) for supplying multiple odors, where the working principles of the OGs variate from the piezoelectric based atomizer¹⁶ to the controllable phase change of the odorous wax¹⁷. However, both two types of OGs require second-level response time, resulting in an obvious latency for users during the practical applications, where the response time of the OGs is defined as the delay time in generating or terminating odors. For example, Metaverse, as one of the most important olfactory interface application fields, is demanded to provide a smooth, immediate, and immersive experience to users, requiring an extremely low delay time from both human machine interface system and network.”

On Page 9 Line 11, we have modified the text as following “To assess the response time of OGs in controlling odors, we adopted a commercial ethanol sensor (TGS823, Figaro Engineering Inc.) to respectively monitor the real-time variations of ethanol

concentration with the operation time at the surrounding airflow rates of 0 m/s and 2.76 m/s, during which we closed the breathing hole at 0.33s and 0.72 s (**Supplementary Fig. 8 and Fig. 2i**). It is clear that the both of two ethanol concentrations are increased slightly in an extremely short time (70 ms), then starts dropping down. The concentration enhancements at the time point when the breathing holes are closed are caused by the internal ethanol gas ejected from the OGs, induced by the upward motion of the magnet. As a result, the response time of the OGs could be drastically reduced to 70 ms. Here, it is worth mentioning that closing the breathing holes of the OGs will block the ethanol release from the OGs, further resulting in a significant decrease in ethanol concentration, and higher ambient air flow rates could induce a faster odor diffusion rate, accelerating the decreasing rate of the ethanol concentration (**Supplementary Fig. 8 and Figs. 2h, i**.)”

Fig. S8. Ethanol concentration response along with the operation time, during which the OG is programmed to close the breathing holes at 0.33 s with the ambient wind speed of 0 m/s.

Comment 3: As shown in the Figure 5b, the olfaction interface was mounted onto human forearm. Because there is some distance between olfaction interface and nose, it may take some time to for the OG arrive at the nose. The successful and efficient olfaction interface had better be located near to the nose. The reason why the olfaction interface is located on the arm, and the possibility for miniaturization to be mounted on face near to the nose will provide the readers with the readers.

Our response: We thank the referee for this comment. We agree that the distance between the wearable odor interface and the nose highly affects the performance of the interface, and the mounting of the device on the forearm is more distal compared to the skin region near the mouth. We adopt the forearm as the application area for the wearable olfactory interface by considering the number of integrated OGs, paired power sources, user friendliness, and convenient odorous chemicals refilling.

Number of integrated OGs. The wearable 32-channel olfactory interface is developed to realize the efficient olfactory recovery, therefore, the most critical issue is how to integrate as many OGs as possible in a limited mounted skin area instead of the response time required to transmit the generated odors to human nose as more supplied odors could increase the improvement rate of olfactory function (*Journal of Clinical Medicine* 12, 4761 (2023)). The forearm has a larger covering space for OG array assembly than the skin region near the mouth, and it is a body part that can easily approach the nose while a user can freely move and adjust displacement voluntarily. By mounting the olfactory interface onto the

forearm, the intensity of odors can be determined based on a user's preference by adjusting the distance between the device and user's nose.

Paired power sources. The 32-channel olfactory interface requires a high-power consumption, which could be supplied by the commercial Li-ion battery (10 V, 1500 mAh) or an external DC power source. By mounting the olfactory interface onto the forearm, users could flexibly choose the two types of the power sources according to their preferences in different practical applications.

User friendliness. Here, we developed a mouth-based system by exploiting a facemask to compare the operational efficiency of the forearm and mouth-based interfaces (Fig. 5b, and Supplementary Fig. 21). As a result, distinct drawbacks of the facial mask olfaction interface were observed, especially in the aspect of user-friendliness. First of all, due to the structure of the mask completely covering the mouth and nose, the released odors were not able to be hastily diffused out, and inconvenience to users was induced during respiration, especially during the silence of the olfaction interface. In addition, as shown in Supplementary Fig. 26, the large-area control panel and power sources will require additional skin areas around users' face, furtherly affecting users' head movement.

Odorous chemicals refilling. Lastly, compared to the forearm-based olfaction interface, the difficulty in refilling odorous chemicals inside OGs increase a lot due to the limitation in wearable formats. As shown in Supplementary Fig. 21c the olfaction interface is entirely embedded in the facial mask, and therefore, to refill the odorous chemicals, the mask must be taken off from the user. Instead, the odorous chemicals can be easily injected into OG units for the forearm-based olfaction interface, even during operation. As a result, we adopted the forearm to mount the olfactory interface.

Although we think the forearm-mounted olfactory interface is more suitable for olfactory training with the reasons illustrated above, the face-mask olfactory interface has unique advantages in VR applications due to the short odor diffusion distance between human nose and the OGs, which is significant in providing users immersive experience in a virtual environment. In the future, we hope to furtherly miniaturize the size of the OGs for providing hundreds of odor options in a small device, which could be mounted onto any body part for operation. We have modified the content and figures for a clear illustration on Page 15 Line 16.

Modification: On Page 15 Line 16, we modified the text as following "It is believed that olfactory training can induce brain neural remodeling and increase the volume of the olfactory bulb, therefore significantly improving the ability of olfactory recognition, discrimination, and the olfactory perception valve³⁰. In addition, some studies have found that increasing the types of odors can increase the improvement rate of olfactory function³¹. Therefore, considering the number of integrated OGs, paired power sources, user friendliness, and convenient odorous chemicals refilling, the forearm is selected as the mounting position due to the large skin surface area and high accessibility to human nose. Based these foundation and remaining challenges, in this new demonstration, a flexible, forearm-mounted olfaction interface with 32 OGs embedded is introduced, paired with a self-developed, flexible control panel connected by four bendable flat cables (**Figs. 5a, b**). Compared to the face mask based olfactory interface with 12 OGs integrated, the forearm-mounted olfactory interface supplies more odor options in a more user-friendly method for a higher efficiency in conducting the olfactory training (**Fig. 5b and Supplementary Fig. 21**). Here, the gap between OGs is optimized by investigating the permanent magnetic force effect on the motion of the magnets inside the OGs (**Supplementary Fig. 22**), and it is found that the magnets of OGs could operate as

normal once the distance between the center of the magnets is over 1.5 cm. In addition, the corner-to-corner OG layout (**Supplementary Fig. 22c**) could contribute to a higher OGs array density (0.75 units/cm^2) than that (0.55 units/cm^2) of the side-by-side layout (**Supplementary Figs. 22a, b**), as observed in **Figs. 5a, b** and **Supplementary Fig. 21**. Therefore, the 32-channel olfactory interface adopts the corner-to-corner OGs layout with a height gap of 2 mm (**Supplementary Figs. 22c, d, e**).".

Fig. S21. a, b Optical images of the 4×3 OGs array mounted onto a self-designed FPCB with the top and bottom views. c Optical image of the face-mask olfactory interface based on the 4×3 OGs array.

Comment 4: The OGs adopts a solenoid valve structure to majorly control the odor release, where the breathing holes at the top PET layer could be opened and closed by manipulating the electromagnetic field. The olfactory interface supports 32 different common odor types, including fruits, spices, drinks, bakeries, herbs, and others. To minimize the unpredictable artifacts on the olfaction interface on these realistic models, all wire connections are hidden behind the OGs mounting surfaces. I wonder how 32 different odor was selected and if 32 odor types can generate any arbitrary smells.

Our response: We thank the referee for this comment. We selected the 32 different odors based on a principle of easy access and biocompatibility, which means that anyone could purchase the 32 different odorous chemicals easily in the market without any special qualifications or permissions. In addition, the 32 odors are very common in the daily life, therefore, in the subsequent volunteer tests (**Figs. 5, 6**), tens of volunteers do not need to spend much time remembering the names of the 32 odors during the olfactory training, which could greatly improve the efficiency of olfactory training.

To investigate if 32 odor types could generate any arbitrary odors, we conducted a new volunteer test, where 9 different odors are selected randomly from the 32 odors for simplifying the volunteer test (see the odor details in **Supplementary Fig. 30**). Among the 9 adopted odors, any 2 different odors are generated by the olfactory interface to create 36 different odor mixtures. During the volunteer test, we recorded the possibility if the volunteers could sense a completely new odor after two different odors were generated, and it is found that the possibility varies from 0.19 to 0.64, demonstrating that our olfactory interface can not stably provide a completely new odor to users by blending the odors generated by the device, which may result from some parameters, including the inconsistent distances between 32 OGs and human nose, distinguished odor diffusion rates, and unpredictable ambient air flow rate. As a result, we can conclude two facts: (1) blending two different odors could create a completely new odor to users, but we are not sure if the 32 odors could generate any arbitrary smells as there are countless odor combinations when considering the mixed odor

number, odor types, and odors concentration ratio among the mixture. (2) Our olfactory interface cannot stably generate a completely new odor, therefore, in the demonstration of the olfactory training, the olfactory interface is programmed to generate one single odor at one time for ensuring the training efficiency. We have modified the content and figures for a clear illustration on Page 16 Line 5.

Modification: On Page 16 Line 5, we have modified the text as following “**Supplementary Fig. 30** demonstrates a volunteer test (6 males and 4 females) to investigate if any two different odors generated by the 32-channel olfactory interface could be blended into a new odor for users, where 9 different odors (see details in **Supplementary Fig. 30**) are randomly selected to create 36 different odor combinations for evaluating the possibility in sensing a new odor. It is obvious that the possibility ranges from 0.19 to 0.64, illustrating a fact that the newly developed high-channel olfactory interface could not stably provide new odors to users by blending the odors embedded as there are many uncertain parameters affecting the mixture of generated odors, for example, the inconsistent distances between 32 OGs and human nose, distinguished odor diffusion rates, and unpredictable ambient air flow rate. Therefore, the 32-channel olfactory interface is programmed to generate one single odor at one time for efficient olfactory training.”

Fig. S30. A volunteer test demonstrating the possibility for volunteers in sensing a new odor when two different odors are generated by the 32-channel olfactory interface at one time. Here, we randomly selected 9 different odors to create 36 (C_9^2) different odor combinations for the volunteer test, including strawberry, cake, ginger, lemon, rice, tobacco, green tea, lavender, and vanilla.

Comment 5: Continuing the previous comments, 32 odor types may be different diffusion rate even at the same temperature. Was each OG calibrated for these kind of diverse diffusion rate depending on the materials?

Our response: We thank the reviewer for the helpful feedback. We agree with the reviewer that the diffusion rate for each odor type is different, and it is also largely affected by environment conditions. For example, in the MR application, we applied the Gaussian plume dispersion model to estimate the spatial-temporal distribution of the odorant concentration. Governed by the advection-diffusion equation, the Gaussian plume model describes the combination of turbulent diffusion and advection due to the air flow as in equation 4,

$$C(x, y, z, h) = \frac{NQ}{2\pi V \sigma_y \sigma_z} \exp\left(\frac{-y^2}{2\sigma_y^2}\right) \cdot \left(\exp\left(\frac{-(z-h)^2}{2\sigma_z^2}\right) + \exp\left(\frac{-(z+h)^2}{2\sigma_z^2}\right)\right)$$

where x and V represent the downwind direction and the average wind/airflow speed, and y stands for the crosswind axis. Q represents the source strength (g/s) of each OG which relates to the discharge hole area, temperature, gas density, upstream pressure, ambient pressure, and the nature of each odorant such as molecular weight. We measured the average Q as the mass of 0.04 ml odor sources divided by the full-load operation time for each OG. Since the diffusivity perfume molecules (0.6 to 1.3 mm²/s, (*In Advances in Material Science and Engineering: Selected articles from ICMMPPE 2020 (pp. 402-409). Springer Singapore.*)) is much slower than those that travel through air currents (0.92 m/s) or convections (avg. 0.27 m/s) in the MR application, we followed the Gaussian plume model assumption that the diffusion term in downwind axis x is neglectable, and the eddy diffusivities in crosswind axes are functions of the downwind distance x only. Then, based on the properties of the airflow, we determined the diffusion coefficients in crosswind axes according to the Briggs expression table. We have modified the content and figures for a clear illustration on Page 12 Line 8.

Modification: On Page 12 Line 8, we have modified the text as following “Specific to our application, we only consider the average odor concentration when diffusion dominates the flow dynamics in crosswind direction with slow laminar flows. To further simplify the model, we neglected the turbulent airflow and evaluated the time-averaged plume envelope in the odor dispersion process. The odor concentration gradients are calculated using the Gaussian plume dispersion model²⁶, considering a working OG as a constant release source. In contrast, diffusion coefficients are calculated according to Briggs expression table²⁷ and atmospheric stability Pasquill classes²⁸. (See more details in Materials and Methods).”

On Page 24 Line 11, we have modified the text as following “**Odor landscape simulation.** The spatial-temporal distribution of odorants within airflow is determined by the fluid dynamics of the ambient atmosphere (with the Navier-Stoke equation) and the motion of the odorant within it. The equation that governs the dynamics of the odorant concentration inside the airflow is the advection-diffusion equation, as advection refers to bulk motion, and diffusion refers to Brownian motion. The balance between these two processes is described by the Péclet number⁴⁰, which is the ratio between the rate of advection and the rate of diffusion. Specific to our application, we only consider the average odor concentration when diffusion dominates the flow dynamics in crosswind direction with slow laminar flows. We calculated the spatial-temporal plume generated by N OGs with Gaussian dispersion plume model:

$$C(x, y, z, h) = \frac{NQ}{2\pi V \sigma_y \sigma_z} \exp\left(\frac{-y^2}{2\sigma_y^2}\right) \cdot \left(\exp\left(\frac{-(z-h)^2}{2\sigma_z^2}\right) + \exp\left(\frac{-(z+h)^2}{2\sigma_z^2}\right)\right) \quad \text{Equation (4)}$$

where $C(x, y, z, h)$ is the average concentration of diffusion substance at a point (x, y, z) in OG’s coordinate system, x and V represent the downwind direction and the average wind/airflow speed, and y stands for the crosswind axis. As for the case of OS 2 with slow horizontal wind, the last term model diffusions along vertical axis z and h are the effective stack height of the odor source. The width of the plume is governed by the rate of turbulent diffusion and is expressed in terms of the dispersion coefficients in crosswind axes.”

Comment 6: The authors only consider the average odor concentration when diffusion dominates the flow dynamics. To further simply the model, they neglected the turbulent airflow and evaluated the effect of low-speed laminar flows in the odor diffusion process. The odor concentration gradients are calculated using the Gaussian plume model, considering a working OG as a constant release source. Cause the olfaction interface was mounted onto the

human forearm. The dynamic environment condition needs to be considered because the arms will be usually swinging when the users are walking.

Our response: We thank the reviewer for this helpful comment. In the application of the MR system, the OGs are mounted on real objects (dummy fruits, flowers, drinks and food), and these odor sources emit odorants in the air that, most of the time, is turbulent and form highly complex odor plumes. We estimated the spatial-temporal distribution of odor plumes using Gaussian dispersion model considering both the physical properties of the airflow and the odorants. For simplicity, the Gaussian plume predicted the time-averaged plume envelope over the turbulent airflow instead of an instantaneous plume eddy boundary which may require large computational cost.

In the application of olfactory training, the 32-channel olfactory interface is mounted onto human forearm. It is worth mentioning that in this demonstration, we didn't integrate the latency-free AI algorithm with the wearable olfactory interface due to a fact that during the olfactory training, one odor generated by the interface require the minute-scale recovery time (*Nature Communications 2023, 14, 1-14*) to fully dissipate in the air. As all the volunteers were required to smell a new generated odor after the former odor fully dissipates in the air for avoiding mixing the two odors together, it is meaningless to integrate the latency-free AI algorithm with the wearable olfactory interface. Therefore, we don't consider the dynamic environment condition when utilize the wearable olfactory interface for olfactory training. We have modified the content and figures for a clear illustration on Page 12 Line 8.

Modification: On Page 12 Line 8, we have modified the text as following “Specific to our application, we only consider the average odor concentration when diffusion dominates the flow dynamics in crosswind direction with slow laminar flows. To further simplify the model, we neglected the turbulent airflow and evaluated the time-averaged plume envelope in the odor dispersion process. The odor concentration gradients are calculated using the Gaussian plume dispersion model²⁶, considering a working OG as a constant release source. In contrast, diffusion coefficients are calculated according to Briggs expression table²⁷ and atmospheric stability Pasquill classes²⁸. (See more details in Materials and Methods).”

On Page 24 Line 11, we have modified the text as following “**Odor landscape simulation.** The spatial-temporal distribution of odorants within airflow is determined by the fluid dynamics of the ambient atmosphere (with the Navier-Stoke equation) and the motion of the odorant within it. The equation that governs the dynamics of the odorant concentration inside the airflow is the advection-diffusion equation, as advection refers to bulk motion, and diffusion refers to Brownian motion. The balance between these two processes is described by the Péclet number⁴⁰, which is the ratio between the rate of advection and the rate of diffusion. Specific to our application, we only consider the average odor concentration when diffusion dominates the flow dynamics in crosswind direction with slow laminar flows. We calculated the spatial-temporal plume generated by N OGs with Gaussian dispersion plume model:

$$C(x, y, z, h) = \frac{NQ}{2\pi V \sigma_y \sigma_z} \exp\left(\frac{-y^2}{2\sigma_y^2}\right) \cdot \left(\exp\left(\frac{-(z-h)^2}{2\sigma_z^2}\right) + \exp\left(\frac{-(z+h)^2}{2\sigma_z^2}\right)\right) \quad \text{Equation (4)}$$

where $C(x,y,z,h)$ is the average concentration of diffusion substance at a point (x, y, z) in OG's coordinate system, x and V represent the downwind direction and the average wind/airflow speed, and y stands for the crosswind axis. As for the case of OS 2 with slow horizontal wind, the last term model diffusions along vertical axis z and h are the effective

stack height of the odor source. The width of the plume is governed by the rate of turbulent diffusion and is expressed in terms of the dispersion coefficients in crosswind axes.”

Comment 7: Unlike visual, auditory, and tactile sensory channels, olfaction is nonlinear chemical sense, making it difficult to develop a technically comprehensive olfaction feedback system in precisely controlling odors generation and delivery. This study directly starts the discussion of olfactory interface without discussing general various other haptic feedbacks for artificial feeling generation (Nat. Rev. Mater., 7, 841, 2022); Adv. Funct. Mater., 31, 2106546 (2021)) including mechano-haptic (mechanical artificial feeling, Nature, 575, 473 (2019); Adv. Funct. Mater., 31, 2007772, (2021)), thermo-haptic (thermal artificial feeling, Adv. Funct. Mater., 30, 1909171, (2020); J. Mater. Chem. A, 8, 8281 (2020); Adv. Funct. Mater., 31, 2007376, (2021)). They need to be briefly discussed in the introduction part to provide the readers with the related recent progresses.

Our response: We thank the referee for this comment. We agree that these references are useful in clarifying our research background. We have modified our manuscript accordingly on Page 3 Line 2.

Modification: On Page 3 Line 2, we have modified the text as following “Olfaction, as an ancient evolutionarily critical physiologic system in humans, plays a significant role in our interaction with the surroundings, including but not limited to detecting potential hazards in the environment, shaping social behavior, smoothing negative emotions, and recalling buried memories¹. Despite the enormous influence of olfaction on human life, a handful of olfaction generating technologies (also called olfactory interfaces) have been developed for olfactory display in some potential applications, ranging from human-machine interactions, to education, to entertainment, and to clinical treatment²⁻⁴. Compared to the extensive research works on the development of haptic feedback technologies⁵⁻¹³, olfactory generating technologies obtain little attention in the recent years, resulting in long-term technical stagnation. Considering the natural defects of massive olfaction generators in overall size, response time and power consumption, the development trend of next-generation olfactory interfaces is approaching to wearable formats with miniaturized design to achieve rapid olfaction feedback and generate localized odorous environment^{2,14,15}.”

The newly added reference:

[7] Oh, J. et al. A liquid metal based multimodal sensor and haptic feedback device for thermal and tactile sensation generation in virtual reality. *Advanced Functional Materials* 31, 2007772 (2021).

[9] Kim, D. et al. Highly stretchable and oxidation-resistive Cu nanowire heater for replication of the feeling of heat in a virtual world. *Journal of materials chemistry A* 8, 8281-8291 (2020).

[10] Lee, J., Kim, D., Sul, H. & Ko, S. H. Thermo-haptic materials and devices for wearable virtual and augmented reality. *Advanced Functional Materials* 31, 2007376 (2021).

[11] Pyun, K. R., Rogers, J. A. & Ko, S. H. Materials and devices for immersive virtual reality. *Nature Reviews Materials* 7, 841-843 (2022).

[13] Lee, J. et al. Stretchable skin-like cooling/heating device for reconstruction of artificial thermal sensation in virtual reality. *Advanced Functional Materials* 30, 1909171 (2020).

[14] Xu, S., Jayaraman, A. & Rogers, J. A. Skin sensors are the future of health care. *Nature* 571, 319-321 (2019).

Comment 8: For the accurate OG control, the released chemical should be monitored and the active feedback to the controllers. I wonder if current system uses active control system or passive control system to accurately control the released amount of OG.

Our response: We thank the referee for this comment. We strongly agree that to increase the OG control accuracy, the released amount of chemicals from each OG unit should be monitored. However, considering the miniaturized size of the OG unit, the variety of the adopted perfume chemicals, and unavailable odor sensors to these adopted perfumes, the installation of the odor sensors inside each OG is not achievable in the current stage. For example, the number of chemical components that the available odor sensors can detect is immensely limited, like smoke, methane, butane, benzene, alcohol, liquefied natural gas (LNG), and hydrogen, which is insufficient to cover the range of chemicals adopted in our olfaction interface system. Therefore, it is clear that discovering an optimal method for rapidly and accurately monitoring the released amount of specific chemical components is still a big challenge in this research field.

For our olfaction interface, the odors releasing rate can be controlled actively by exploiting the heating electrode and solenoid valve embedded in the OG unit. The OGs could accurately control the heating temperature and duty cycle of the AC power into the electromagnetic coil for determining the odor generation rate, where higher heating temperature and lower duty cycle could contribute to a higher odor generation rate (Figs. 2c, g, and Movies S4, 5). We have modified the content and figures for a clear illustration on Page 4 Line 4.

Modification: On Page 4 Line 4, we have modified the text as following “This design could significantly improve the mechanical stability of the OG, especially when users do intense activities; a layer of cotton/adhesive (0.1 mm in thickness) works as the odorous chemicals container, where the chemicals could be in either liquid or powders formats; a layer of the polyimide (PI, 2 μm thick), a metallic electrode composed of gold (Au, 200 nm), chromium (Cr, 40 nm), and a tiny thermistor (1 mm \times 0.5 mm \times 0.5 mm) act as a heating platform with controllable operation temperature (from 35 to 55°C, shown in **Supplementary Fig. 4**), therefore, controlling the odor generation rate; a 3-dimension (3D) printed square ring (epoxy) supports the whole the structure against the unpredictable external impact. The OG adopts a solenoid valve structure to majorly control the odor release rate, where the breathing holes at the top PET layer could be opened and closed by manipulating the electromagnetic field (**Figs. 1a, b, c**). For operating the OGs, an external powering management system provides two channels of voltage inputs (1.7 V and 9.0 V) to respectively supply the electromagnetic coil and heating electrode for controlling the magnet motion and heating temperature insides. To monitor the heating temperature, the miniaturized thermistor is connected to an Analog-to-Digital Converter and General-purpose input/outputs of the microcontroller in a self-developed control panel for continuously measuring the temperature (sensing resistance) variation during the operation. According to the real-time heating temperature, the control panel could rapidly turn on/off the power supply into the heating electrode for stabilizing the temperature at the target one. To change the duty cycles, amplitude and frequency of the alternating current (AC) power into the electromagnetic coil, the control panel adopts the

decoder controlled by two digital General-purpose input/outputs to convert external direct current (DC) power source to AC power for realizing a programmable magnetic up-and-down displacement (**Supplementary Notes 1, 2**).”

Comment 9: Some of the digital figures are missing scale bars. They should be added to the pictures.

Our response: We thank the referee for this comment. We have added the scale bars accordingly in Figs. 1, 4, 6 and Supplementary Figs. 4, 8.

Modification: We have modified the Figs. 1, 4, 6 accordingly as following:

Fig. 1. Concept of the newly developed olfaction interfaces. a) Exploded view of the odor generator with detailed descriptions on each component. b) Optical images and schematic diagram of the OG with the additional inner views. c) Manipulation methods of the OGs in controlling odor generation rate. d, e) Mechanical stimulations of the Au based heating electrode (d) and cantilever-like PET film (e) under the distributed external pressure and bending deformation, respectively. f) Comparison of the new OGs and the recently reported ones in terms of response time, lowest power consumption in operating OGs, overall size, stability, operation temperature, and the maximum number of supply odors. g) Schematic diagram of the advanced OGs based olfaction interfaces for providing olfaction feedbacks to users at four typical applications, ranging from MR based entertainment and education to olfaction training for olfactory recovery and enhancement.

Fig. S4. Thermal distribution of the OGs insides with the target temperatures varying from 35°C to 55°C. Here, due to the fully encapsulated structure design, we remove the PET cantilever and breathing layers for exposing the inner heating parts contributed from both the copper coil and the heating electrode.

Fig. 4. One application of the MR system with the olfaction interfaces integrated. a) Optical images of the whole MR system in reality and virtuality, where a volunteer is walking around the four odor sources for randomly enjoying the target released odors. **b)** the volunteer's exploration trajectory and facing directions within the odor landscape. **c)** Plotting of the odor plumes of OS 1 shown in **b)**. **d)** Distance from the user's nose towards the downwind axis of each odor sources, $|\overline{UO_p}|$. **e)** Visualization of command signals sent to the corresponding newly developed olfaction interfaces along the volunteer's trajectory. **f)**

Comparison between the potential distance from odor plumes to the volunteer and the exploration potential, $\widehat{D}_{OpU} - \widehat{D}_{UOp}$. **g)** Command signals sent to the corresponding olfaction interfaces during volunteer's exploration for achieving a latency-free experience with minimal energy and odorous sources consumption. **h)** Stimulation results showing the command signals sent to the corresponding previously reported olfaction interfaces with the response time of 1.4 s along the same volunteer's trajectory. **i)** The stimulated comparison between the potential distance from odor plumes to the volunteer and the exploration potential, $\widehat{D}_{OpU} - \widehat{D}_{UOp}$, for the reported olfaction interfaces. **j)** The stimulated command signals sent to the reported olfaction interfaces during volunteer's exploration for achieving a latency-free experience with minimal energy and odorous sources consumption.

Fig. S13. Optical image of the fake pancake dish with 10 OGs integrated for generating different food odor types.

Fig. 6. One typical application of the AI-driven olfaction interface in olfaction enhancement. **a)** The adopted list-wise recommendation frame for providing a personalized olfaction training plan to users. **b)** Olfactory capability variations of 10 volunteers without any olfactory impairment after receiving 7-day personalized olfaction training achieved by the intelligent, high-channel olfaction interface system. **c, d)** Overall recognition rate (c) and training time (d) variations of the 10 volunteers as a function of olfaction training times.

Comment 10: The general usage of language including the typos and grammatical errors need to be checked again.

Our response: We thank the referee for this comment. We have revised the manuscript throughout as following:

Modification: On Page 4 Line 7, we have modified the text as following “Here, we report a series of materials, algorithms, devices, mechanics, electronics and integration strategies for AI-driven, wireless olfactory interfaces, which exhibit world-record high response time (0.07 s), OG array density (0.75 unit/cm²) and latency-free feature. The miniaturized OGs used in the olfactory array adopt air flow and heat as odor-generating factors as working principle, where the release of odors is controlled by a mechanical actuator to open/close the breathing holes in the OG, while the odor concentration is controlled by the tuning of heating temperature (**Fig. 1a and Supplementary Figs. 1 and 2**). The combination of advanced AI algorithms with the olfactory interface allows zero latency for the olfactory interface in mixed reality (MR) applications, which enables efficient olfaction training for patients who are experiencing the olfactory degeneration.”

On Page 6 Line 3, we have modified the text as following “**Figs. 1d, and Movies S1** show that the simulated equivalent strains in the electrode remain lower than the elastic limit (0.3%) in the external distributed pressure of 8.8 kPa.”

On Page 6 Line 11, we have modified the text as following “The core technical parameters of the OG reported in this work show significant improvement, compared to the state-of-the-art ones, including the response time in controlling odor release, the lowest operating power consumption, overall size, stability of repeatedly switching the heating temperatures, minimum operation temperature, and maximum number of available odor types in one olfaction interface (**Fig. 1f**)^{2,4}. Due to the excellent performances of the new OG, the olfaction interfaces based on it can be utilized in various scenarios ranging from MR based entertainment, education, olfaction recovery, and enhancement (**Fig. 1g**).”

On Page 7 Line 22, we have modified the text as following “Considering the available commercial ethanol sensor (TGS823, Figaro Engineering Inc.) to detect the odor concertation, we chose the ethanol solution (vol%, 95%) as the odorous chemical stored inside OGs instead of other biosafe odorous chemicals.”

On Page 8 Line 9, we have modified the text as following “It is obvious that the peak ethanol concentrations at the three frequencies are stabilized around 2717 ppm with minor differences.”

On Page 11 Line 19, we have modified the text as following “Here, the VR glasses are programmed to continuously measure the users’ walking velocity (V_u), facing direction, and 3D location (x_u, y_u, z_u).”

On Page 13 Line 6, we have modified the text as following “The flexible format of the control panel allows it to mount on human skin for long-term wearability, where users can also wear it in the necklace and ring models during MR applications.”

On Page 13 Line 17, we have modified the text as following “As the full-load operation time of the 32 adopted odors ranges from 0.26 to 2.7 hrs on average at a constant volume of 0.04 mL (**Fig. 3f**), the 500 mAh battery is sufficient for the operation of the olfaction systems in various MR tasks.”

On Page 14 Line 7, we have modified the text as following “**Fig. 4 and Movie S3** demonstrate a typical application of the above-mentioned MR system with no-delay olfaction feedback, where the four odor sources (OSs) in reality share the same 3D locations as those in virtuality by utilizing the positioning system (**Fig. 4a**).”

On Page 18 Line 10, we have modified the text as following “Benefitted from the miniaturized size and ultra-low response time (70 ms) of the OGs, the flexible, high-

channel olfactory interface could be directly mounted onto volunteers' forearms (**Fig. 5b**) and rapidly switch 32 different odor types with imperceptible delay."

On Page 19 Line 2, we have modified the text as following "Lower scores of both forms correspond to more positive emotions. The results show that the value of HADS depression (HADS D) variated a little from 3.1 to 3.2 after olfactory training, but values of HADS A and PHQ-9 changed significantly from 5.5 and 3.2 to 5.2 and 2.3, indicating that the volunteers became more positive after 3 olfaction training sessions."

REVIEWERS' COMMENTS

Reviewer #1 (Remarks to the Author):

I am satisfied by the authors' reply.

Reviewer #2 (Remarks to the Author):

The current manuscript is with good quality to be accepted for publication in Nature Communications.

Reviewer #2 (Remarks on code availability):

The readme file is provided with enough instructions, and the code will be useful for future works in multiple channels processing.

Reviewer #3 (Remarks to the Author):

The authors responded well to the comments.

Responses to comments of Referee #1

Comments from Referee #1:

Summary Comment: I am satisfied by the authors' reply.

Our response: We thank the referee for reviewing our work.

Modifications: None

Responses to comments of Referee #2

Comments from Referee #2:

Summary Comment: The current manuscript is with good quality to be accepted for publication in Nature Communications. The readme file is provided with enough instructions, and the code will be useful for future works in multiple channels processing.

Our response: We thank the referee for reviewing our work.

Modifications: None

Responses to comments of Referee #3

Comments from Referee #3:

Summary Comment: The authors responded well to the comments.

Our response: We thank the referee for reviewing our work.

Modifications: None

Description of Additional Supplementary Movies

Video name: Movie S1

Description: Simulated equivalent strains in the Au electrode in the external distributed pressure of 8.8 kPa.

Video name: Movie S2

Description: Simulated equivalent strains in the cantilever-like PET film in the OGs when the magnet is driven by the uniformly distributed load to bend the PET film downwards by 29°.

Video name: Movie S3

Description: This movie demonstrates a typical application of the MR system with latency-free olfaction feedback, where the four odor sources (OSs) in reality share the same 3D locations as those in virtuality by utilizing the positioning system. During the demonstration, a male volunteer walked randomly around the four OSs for enjoying the released odors by target OSs with the routine and facing direction. As a result, the algorithm evaluates the differences between the potential exploration intents and odor landscapes and ensures when the volunteer arrives at the edge of the odor plumes, the odor concentration rate has reached at 5 ppm just in time. By demanding OGs to work at the temporal-spatial crossing point of these two potentials, the algorithm achieves latency-free experiences with minimal electrical energy waste.

Video name: Movie S4

Description: This movie demonstrates that we could teleoperate the wearable, 32-channel olfactory interface to separately generate a single odor among the 32 different odor types.

Video name: Movie S5

Description: This movie demonstrates that we could teleoperate the wearable, 32-

channel olfactory interface to control the odor release rate by adjusting the heating temperature and duty cycle of AC power into the electromagnetic coil of the OGs.